# USB-NeRF: Unrolling Shutter Bundle Adjusted Neural Radiance Fields

**Moyang Li**[1,2*]    **Peng Wang**[1,3*]    **Lingzhe Zhao**[1]    **Bangyan Liao**[1,3]    **Peidong Liu**[1†]

[1]Westlake University      [2]ETH Zürich      [3]Zhejiang University

moyali@ethz.ch, {wangpeng, zhaolingzhe, liaobangyan, liupeidong}@westlake.edu.cn

## ABSTRACT

Neural Radiance Fields (NeRF) has received much attention recently due to its impressive capability to represent 3D scene and synthesize novel view images. Existing works usually assume that the input images are captured by a global shutter camera. Thus, rolling shutter (RS) images cannot be trivially applied to an off-the-shelf NeRF algorithm for novel view synthesis. Rolling shutter effect would also affect the accuracy of the camera pose estimation (e.g. via COLMAP), which further prevents the success of NeRF algorithm with RS images. In this paper, we propose Unrolling Shutter Bundle Adjusted Neural Radiance Fields (USB-NeRF). USB-NeRF is able to correct rolling shutter distortions and recover accurate camera motion trajectory simultaneously under the framework of NeRF, by modeling the physical image formation process of a RS camera. Experimental results demonstrate that USB-NeRF achieves better performance compared to prior works, in terms of RS effect removal, novel view image synthesis as well as camera motion estimation. Furthermore, our algorithm can also be used to recover high-fidelity high frame-rate global shutter video from a sequence of RS images. Code and data are available at https://github.com/WU-CVGL/USB-NeRF.

## 1 INTRODUCTION

Understanding and recovering 3D scenes from 2D images is a difficult but important problem in computer vision. Different from a 2D image which can be naturally formulated as an array of pixel values, there are many 3D representations to depict a 3D scene, such as the commonly used point clouds (Furukawa & Ponce, 2009), height-map (Pollefeys et al., 2008), voxel grids (Nießner et al., 2013; Seitz & Dyer, 1997) and 3D triangular meshes (Delaunoy & Pollefeys, 2014). Each has its own advantages and limitations. Recently, implicit neural representation by Neural Radiance Fields (NeRF) (Mildenhall et al., 2020) has drawn great attention, due to its impressive 3D representation capability. NeRF represents the scene with a Multi-layer Perception (MLP) network. It takes a 5D vector (i.e. the 3D position and 2D viewing direction of a query point) as input and outputs the corresponding radiance and volume density of the query point. The pixel intensity is then accumulated by differentiable volume rendering (Levoy, 1990; Max, 1995). The parameters of the MLP network can be estimated by maximizing the photo-metric consistency across images captured from different viewpoints.

To obtain a good representation of the 3D scene with NeRF, both high-quality images and corresponding accurate camera poses are usually necessary. However, it is usually difficult to acquire such perfect inputs in real-world scenarios, as real images can be easily degraded by motion blur, de-focus, rolling shutter (RS) effect etc. Different from the commonly assumed global shutter camera model by NeRF, rolling shutter cameras capture images row by row sequentially, as illustrated in Figure 3. Different rows of the image are thus scanned at different timestamps, which would lead to rolling shutter distortions if it is captured by a moving camera. Neglecting these distortions usually can lead to performance degradation in 3D reconstruction, motion estimation as well as camera localization etc., via rolling shutter images. A trivial solution to mitigate the effect of the rolling shutter distortions is to apply a state-of-the-art RS effect correction algorithm (Liu et al., 2020; Fan & Dai, 2021; Fan et al., 2022) to pre-process the images before they are fed into downstream tasks. However, those methods usually require to be pre-trained by a large dataset, which can be expensive to obtain in real-world scenarios. The generalization performance of those pre-trained networks is also limited as demonstrated

---

*Equal contribution.

†Corresponding author.

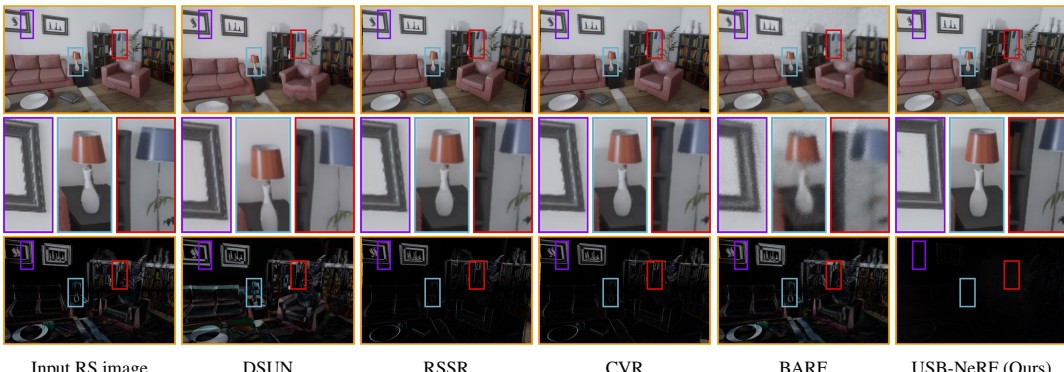

|Input RS image | DSUN | RSSR | CVR | BARF | USB-NeRF (Ours)|

Figure 1: Given a sequence of rolling shutter images, our method is able to simultaneously learn the undistorted 3D scene representation and recover the continuous-time camera motion trajectory. Global shutter images with removed rolling shutter effect can then be rendered from the learned 3D representation. The **third row** presents residual images (the darker the better) that are defined as the absolute difference between the corresponding images (**first row**) and ground truth global shutter images.

in our experiments. Therefore, we propose unrolling shutter bundle adjusted neural radiance fields in this paper. The proposed method is able to learn the 3D representation and recover the camera motion trajectory simultaneously. High quality un-distorted global shutter images can be further synthesized (i.e. with RS effect removed) from the learned 3D scene representation.

In particular, we propose to represent the 3D scene with NeRF and model the camera motion trajectory with a differentiable continuous-time cubic B-Spline in the $\mathbf{SE}(3)$ space. Given a sequence of rolling shutter images, we aim to optimize the camera motion trajectory (i.e. estimate the parameters of the cubic B-Splines) and learn the implicit 3D representation simultaneously. The optimization is achieved by formulating the real physical image formation process of an RS camera, and maximizing the photometric consistency between the rendered and captured RS images. The method thus does not require any pre-training and would have better generalization performance compared to prior learning-based works as demonstrated in our experiments. Given the estimated continuous-time motion trajectory and the learned 3D scene representation, we can further recover the global shutter images in arbitrary desired frame-rate in high quality. We dub our method as *USB-NeRF*, i.e., Unrolling Shutter Bundle Adjusted Neural Radiance Fields.

Extensive experimental evaluations are conducted with both synthetic and real datasets, to evaluate the performance of our method. The experimental results demonstrate that *USB-NeRF* achieves superior performance compared to prior state-of-the-art methods (e.g. as shown in Figure 1) in terms of rolling shutter effect removal, novel view image synthesis as well as camera motion estimation.

## 2 RELATED WORK

We review the related works in two main areas: neural radiance fields and rolling shutter effect correction, which are the most related to our work.

**Neural Radiance Fields.** NeRF has received lots of attention recently due to its impressive capability to represent 3D scenes (Mildenhall et al. (2020)). Many extensions have been proposed to further improve its performance. For example, Müller et al. (2022); Yu et al. (2021a); Fridovich-Keil et al. (2022); Chen et al. (2022a); Garbin et al. (2021) proposed approaches to accelerate its training and rendering efficiency. Other extensions also explore NeRF for dynamic scenes (Pumarola et al. (2021); Gao et al. (2021); Park et al. (2021); Tretschk et al. (2021)) and scene editing (Li et al. (2022); Liu et al. (2021); Sun et al. (2022); Kania et al. (2022)). Apart from this, there are also many variants have been proposed to address the training of NeRF with imperfect inputs, such as with unknown or inaccurate poses (Wang et al. (2021); Lin et al. (2021); Chen et al. (2022b); Meng et al. (2021)), with degraded images (e.g. blurry images Ma et al. (2022); Wang et al. (2023), dark/noisy images Mildenhall et al. (2022); Pearl et al. (2022), low dynamic range images Huang et al. (2022)), or with a limited number of input images etc. (Niemeyer et al. (2022); Yu et al. (2021b); Kim et al. (2022); Xu et al. (2022); Deng et al. (2022)).

We will review those methods in detail which are the most related to our work as follows. To overcome the effect of inaccurate camera poses, NeRF-- (Wang et al. (2021)) sets the camera poses as learnable parameters and optimizes them with the weights of NeRF jointly by minimizing the photo-metric loss.

GNeRF (Meng et al. (2021)) further integrates an additional adversarial loss into the training of the whole pipeline to have a better camera pose estimation. BARF (Lin et al. (2021)) and L2G-NeRF (Chen et al. (2022b)) propose to gradually apply the positional encoding to achieve a coarse-to-fine training strategy, to better constrain the training of the network and camera pose estimation. Although these methods have achieved impressive results with imperfect poses, images with rolling shutter effect are still a problem for NeRF. Prior works usually assume a global shutter camera model and use a single transformation matrix to represent the pose of each view. They are thus not suitable for rolling shutter camera model, in which each row has different poses. We therefore parameterize the whole motion trajectory of the RS image sequence with a differentiable continuous-time cubic B-Spline parameterized in the $\mathbf{SE}(3)$ space. We then formulate the image formation process of a rolling shutter camera into the joint training of NeRF and the parameter estimations of the cubic B-Splines.

**Rolling Shutter Effect Correction.** RS effect removal is a challenging problem, and many related methods have been proposed over the last decades Forssen & Ringaby (2010); Baker et al. (2010); Rengarajan et al. (2016); Purkait et al. (2017); Lao & Ait-Aider (2018); Vasu et al. (2018) etc. We will detail several recent state-of-the-art methods as follows. Hedborg et al. (2012) propose to recover the camera poses and sparse 3D geometry from a sequence of rolling shutter images. They assume a piece-wise linear motion model for each frame and propose a sparse bundle adjustment solver for rolling shutter cameras. Grundmann et al. (2012) present a mixture model of homographies to model rolling shutter distortions of video streams. Zhuang et al. (2017) later develop an RS-aware differential Structure from Motion (SfM) algorithm to estimate the relative poses of two consecutive RS images and then rectify the distortions. As for unorganized RS images, RS effect correction has been shown to suffer from severe degeneracy (Albl et al. (2016)). To mitigate the degeneracy of rolling-shutter (RS) SfM, Ito & Okatani (2017) propose to add a critical camera motion constraint; Albl et al. (2020) and Zhong et al. (2022) propose to employ dual RS images with reversed directions to avoid the ambiguity. Deep-learning-based approaches have also been proposed to address RS effect correction recently. Rengarajan et al. (2017) propose a convolutional neural network (CNN) to estimate the row-wise camera motion from a single RS image. Liu et al. (2020) and Fan et al. (2021) design special shutter unrolling networks to recover the global shutter image from two consecutive images. Fan & Dai (2021) and Fan et al. (2022) further developed Acceleration-Net and bilateral motion field approximation model to achieve RS temporal super-resolution. While those methods deliver state-of-the-art performance, they usually require a large dataset for network training. Those datasets are usually expensive to obtain in practice and further limit their generalization performance to images with different characteristics (as shown in our experiments). In contrast, our method does not require any pre-training with large datasets and would thus have no generalization issues.

## 3 METHOD

In this section, we present the details of our unrolling shutter bundle adjusted neural radiance fields (USB-NeRF). USB-NeRF takes a sequence of rolling shutter images as input. It then learns the underlying 3D scene representation and recovers the continuous camera motion trajectory simultaneously, by maximizing the photo-metric consistency between the rendered and captured RS images. The learned 3D representation is free of rolling shutter distortions and thus able to be used for arbitrary frame-rate global shutter image/video synthesis, provided the recovered continuous-time camera motion trajectory. The details of the method are shown in Figure 2. We will detail each component as follows.

### 3.1 NEURAL RADIANCE FIELDS

We represent the 3D scene implicitly with a Multi-layer Perceptron (MLP) network. We adopt the original architecture of NeRF proposed by Mildenhall et al. (2020). More advanced variants of NeRF, such as voxel-based NeRF representation with improved efficiency from Yu et al. (2021a) are also feasible to be used for our method.

Given a camera view with known pose, we can render its corresponding image from the implicit 3D representation by using volume rendering. For convenience, we present the steps to render the intensity of a particular pixel to illustrate the concept. The rendering procedures of other pixels are the same. To render the pixel intensity $\mathbf{I}(\mathbf{x})$ at pixel location $\mathbf{x}$ for a particular image with pose $\mathbf{T}_c^w$, we can query the radiance and volume density of each 3D point along the ray from camera center to the

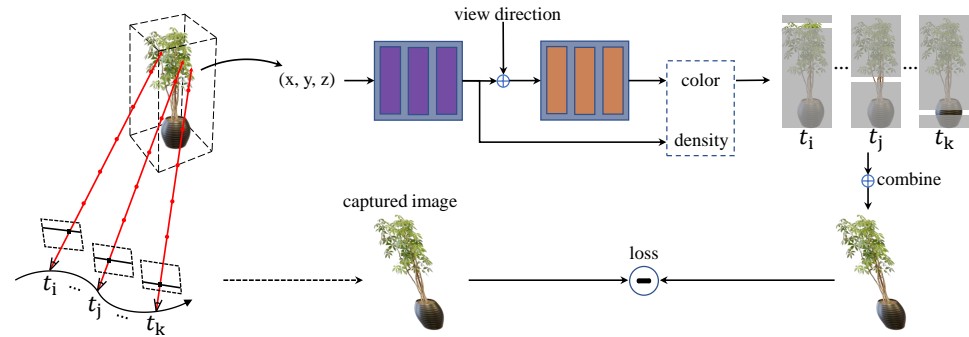

Figure 2: **The pipeline of USB-NeRF.** Given a sequence of rolling shutter images, we train NeRF to learn the underlying undistorted 3D scene representations. We parameterize the camera motion trajectory of the image sequence by a continuous-time cubic B-Spline in $\mathbf{SE}(3)$ space. Given the capturing time for each row of the rolling shutter image, we can interpolate its pose from the spline. Each rolling shutter image can then be synthesized by rendering all the image rows (i.e. each with different poses) from NeRF. By maximizing the photo-metric consistency between the synthesized and captured RS images, we can learn the underlying 3D scene representation and recover the camera motion trajectory. Global shutter images can then be rendered from the learned 3D representation with known camera poses.

3D space passing through $\mathbf{x}$. The pixel intensity (i.e. $\mathbf{I}(\mathbf{x})$) can then be computed by accumulating the sampled radiance and volume densities along the ray. The whole procedure can be formally defined as follows.

Assume the sampled 3D point along the ray has depth $\lambda$, its 3D position $\mathbf{X}^w$ in the world coordinate frame can be computed by (assuming the camera has a pin-hole model):

$$\mathbf{d}^c = \mathbf{K}^{-1} \begin{bmatrix} \mathbf{x} \\ 1 \end{bmatrix}, \tag{1}$$

$$\mathbf{X}^w = \mathbf{T}_c^w \cdot \lambda \mathbf{d}^c, \tag{2}$$

where $\mathbf{d}^c$ is the ray direction defined in the camera coordinate frame, $\mathbf{K}$ is the camera intrinsic matrix, $\mathbf{x}$ is its 2D pixel coordinate, and $\mathbf{T}_c^w$ is the transformation matrix used to convert a 3D vector from the camera coordinate frame to world coordinate frame. We can then query the MLP network $\mathbf{F}_{\boldsymbol{\theta}}$ parameterized by $\boldsymbol{\theta}$ for the radiance $\mathbf{c}$ and volume density $\sigma$ of the sampled 3D point $\mathbf{X}^w$ by:

$$(\mathbf{c}, \sigma) = \mathbf{F}_{\boldsymbol{\theta}}(\gamma_{L_x}(\mathbf{X}^w), \gamma_{L_d}(\mathbf{d}^w)), \tag{3}$$

where $\mathbf{d}^w = \mathbf{R}_c^w \cdot \mathbf{d}^c$ is the viewing direction of the ray defined in the world coordinate frame, $\mathbf{R}_c^w$ is the rotation matrix which transforms vectors from camera frame to world frame, and $\gamma_*$ represents positional encodings for $\mathbf{X}^w$ and $\mathbf{d}^w$ (Mildenhall et al., 2020). The final pixel intensity can then be computed from $N$ sampled 3D points along the ray via:

$$\mathbf{I}(\mathbf{x}) = \sum_{i=1}^{N} T_i(1 - \exp(-\sigma_i \delta_i)) \mathbf{c}_i, \tag{4}$$

where $\mathbf{c}_i$ and $\sigma_i$ are the predicted radiance and volume density of the $i^{th}$ point by Eq. 3 respectively, $\delta_i = \left\| \mathbf{X}_{i+1}^w - \mathbf{X}_i^w \right\|_2$ is the distance between two adjacent points, and $T_i$ represents the transmittance of the $i^{th}$ point and can be computed by:

$$T_i = \exp(-\sum_{k=1}^{i-1} \sigma_k \delta_k). \tag{5}$$

According to the above derivations, we can see that $\mathbf{I}(\mathbf{x})$ is also a function of the camera pose $\mathbf{T}_c^w$. Since the whole rendering procedure is differentiable, $\mathbf{T}_c^w$ can thus also be relaxed as a free parameter to be optimized during the training of the MLP network (i.e. $\mathbf{F}_{\boldsymbol{\theta}}$) (Lin et al., 2021).

## 3.2 ROLLING SHUTTER CAMERA MODEL

Different from global shutter cameras, each scanline/image row of the rolling shutter camera is captured at different timestamps. Without loss of generality, we assume the readout direction of RS camera is from top to bottom as shown in Figure 3 in our formulation. This process can be mathematically modeled as (assuming infinitesimal exposure time):

$$[\mathbf{I}^r(\mathbf{x})]_i = [\mathbf{I}_i^g(\mathbf{x})]_i, \tag{6}$$

where $\mathbf{I}^r(\mathbf{x})$ is the rolling shutter image, $[\mathbf{I}(\mathbf{x})]_i$ denotes an operator which extracts the $i^{th}$ row from image $\mathbf{I}(\mathbf{x})$, $\mathbf{I}_i^g(\mathbf{x})$ is the global shutter image captured at the same pose as the $i^{th}$ row of $\mathbf{I}^r(\mathbf{x})$. We denote the pose of the $i^{th}$ row of $\mathbf{I}^r(\mathbf{x})$ as $\mathbf{T}_{c_i}^w$. Thus, provided the 3D representation by NeRF and the known poses $\mathbf{T}_{c_i}^w$ for $i = 0, 1, ..., (H-1)$, where $H$ is the height of the image, we can easily render the corresponding rolling shutter image $\mathbf{I}^r(\mathbf{x})$.

From the above derivations, we can see that $\mathbf{I}^r(\mathbf{x})$ is a function of $\boldsymbol{\theta}$ (i.e. the weight of the MLP network), and $\mathbf{T}_{c_i}^w$ for $i = 0, 1, ..., (H-1)$. Furthermore, we can also find that $\mathbf{I}^r(\mathbf{x})$ is differentiable with respect to $\mathbf{T}_{c_i}^w$ and $\boldsymbol{\theta}$. It thus lays the foundation for our bundle adjustment formulation with a sequence of rolling shutter images.

### 3.3 CAMERA MOTION TRAJECTORY MODELING

To do bundle adjustment optimization with a sequence of rolling shutter images, we need to parameterize the pose of each row for each image as shown in the previous section. Commonly used parameterization is to assign a 6 DoF pose to the first row of each image and then do linear interpolation for subsequent rows (Hedborg et al., 2012). Instead of using such kind of simple linear motion model, we propose to use cubic B-Splines parameterized in the $\mathbf{SE}(3)$ space in this work, which can handle more realistic complex camera motions (Lovegrove et al., 2013). Experimental ablation studies also verify that cubic B-Splines formulation delivers better performance than the simple linear motion model for complex motions.

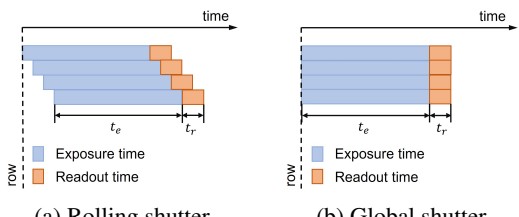

(a) Rolling shutter  (b) Global shutter

Figure 3: **Image formation models of a rolling shutter camera and a global shutter camera respectively.** It demonstrates that each row of a rolling shutter image is captured at different timestamps, and would thus lead to image distortions if the image is captured by a moving camera.

We use a sequence of control knots with poses $\mathbf{T}_{c_i}^w$ * for $i = 0, 1, ..., n$, to represent the spline. For brevity, we denote $\mathbf{T}_{c_i}^w$ with $\mathbf{T}_i$ for subsequent derivations. We assume the control knots are sampled with a uniform time interval $\Delta t$ and the trajectory starts from $t_0$. Spline with a smaller $\Delta t$ can represent a smoother motion, with an expense of more parameters to optimize. Since four consecutive control knots determine the value of the spline curve at a particular timestamp, we can thus compute the starting index of the four control knots for time $t$ by:

$$k = \lfloor \frac{t - t_0}{\Delta t} \rfloor, \tag{7}$$

where $\lfloor * \rfloor$ is the floor operator. Then we can obtain the four control knots responsible for time $t$ as $\mathbf{T}_k$, $\mathbf{T}_{k+1}$, $\mathbf{T}_{k+2}$ and $\mathbf{T}_{k+3}$. We can further define $u = \frac{t - t_0}{\Delta t} - k$, where $u \in [0, 1)$ to transform $t$ into a uniform time representation. Using this time representation and based on the matrix representation for the De Boor-Cox formula (Qin, 1998), we can write the matrix representation of a cumulative basis $\tilde{\mathbf{B}}(u)$ as

$$\tilde{\mathbf{B}}(u) = \mathbf{C} \begin{bmatrix} 1 \\ u \\ u^2 \\ u^3 \end{bmatrix}, \quad \mathbf{C} = \frac{1}{6} \begin{bmatrix} 6 & 0 & 0 & 0 \\ 5 & 3 & -3 & 1 \\ 1 & 3 & 3 & -2 \\ 0 & 0 & 0 & 1 \end{bmatrix}. \tag{8}$$

The pose at time $t$ can then be computed as:

$$\mathbf{T}(u) = \mathbf{T}_k \cdot \prod_{j=0}^{2} \exp(\tilde{\mathbf{B}}(u)_{j+1} \cdot \boldsymbol{\Omega}_{k+j}), \tag{9}$$

where $\tilde{\mathbf{B}}(u)_{j+1}$ denotes the $(j+1)^{th}$ element of the vector $\tilde{\mathbf{B}}(u)$, $\boldsymbol{\Omega}_{k+j} = \log(\mathbf{T}_{k+j}^{-1} \cdot \mathbf{T}_{k+j+1})$.

From the above derivations, we can see that the interpolated camera poses are functions of the poses of the control knots. They are also differentiable with respect to the poses of those control knots.

---

*Here we abuse the same notation as the previously defined transformation matrix of the $i^{th}$ row of the rolling shutter image.

## 3.4 Loss Function

Given a sequence of rolling shutter images, we can then estimate the learnable parameters $\boldsymbol{\theta}$ of NeRF as well as the camera motion trajectory parameterized by cubic B-Spline (i.e. $\mathbf{T}_0$, $\mathbf{T}_1$, ..., and $\mathbf{T}_n$), by minimizing the photo-metric loss:

$$\mathcal{L} = \sum_{m=0}^{M-1} \left\| \mathbf{I}_m^r(\mathbf{x}) - \tilde{\mathbf{I}}_m^r(\mathbf{x}) \right\|_2, \tag{10}$$

where $M$ is the number of input images, $\mathbf{I}_m^r(\mathbf{x})$ is the captured rolling shutter image, and $\tilde{\mathbf{I}}_m^r(\mathbf{x})$ is the rendered rolling shutter image from NeRF, with known camera intrinsic parameters etc. $\tilde{\mathbf{I}}_m^r(\mathbf{x})$ is a function of $\boldsymbol{\theta}$ as well as part of the control knots' poses of the whole trajectory. We implement the above equations with PyTorch and exploit its automatic differentiation module to compute the Jacobian for back-propagation.

## 4 Experiments

### 4.1 Datasets

**Synthetic datasets.** We use the scripts provided by Liu et al. (2020) to synthesize 4 datasets with the Unreal game engine (i.e. Unreal-RS-BlueRoom, Unreal-RS-LivingRoom, Unreal-RS-WhiteRoom, Unreal-RS-Adornment) and 2 datasets with Blender (i.e. Blender-RS-Factory, Blender-RS-Tanabata). We adopt the real motion trajectories from ETH3D (Schops et al., 2019) (i.e. the challenging shaky sequences) to synthesize the images. Since the ground truth pose of ETH3D dataset has only 100 Hz, we further interpolate the trajectories with cubic spline to have a continuous-time representation. We first capture GS images in the Unreal game engine and Blender at the frequency of 10,000 Hz, then synthesize RS images by simulating the physical image formation process of an RS camera. We configure the scanline readout time as 100 $\mu$s and the image resolution as 768×480 pixels. We generate 40 RS images in total for Unreal-RS-BlueRoom, Unreal-RS-LivingRoom, Unreal-RS-WhiteRoom individually and 80 RS images for Unreal-RS-Adornment, Blender-RS-Factory, Blender-RS-Tanabata. We also evaluate our method on a public synthetic dataset (i.e. Carla-RS[†] (Liu et al., 2020)) for fair comparisons against other methods.

**Real datasets.** We captured 5 sequences using GoPro HERO6 Black, Canon camera (EOS M3), and iPhone 14 Pro. All rolling shutter images are captured at the frequency of 30 Hz. The scanline readout time of aforementioned 3 cameras is approximately 13.89 $\mu$s, 18.52 $\mu$s and 3.70 $\mu$s, respectively. We also evaluate our method on the public real-world dataset TUM-RS (Schubert et al., 2019). TUM-RS consists of 10 real challenging indoor sequences of rolling shutter images, which are originally used for RS visual-inertial odometry evaluations. It records RS images, groundtruth motion trajectories at the frequency of 20 Hz and 120 Hz, respectively. The scanline readout time of the RS camera is approximately 29.47 $\mu$s. As whole sequences are too long for NeRF to process, we choose a subset frames from each sequence. Details are presented in the Appendix A.2. Since there are no pixel-aligned RS-global shutter image pairs for this dataset, we only evaluate the accuracy of recovered camera motion trajectories. We also provide additional qualitative comparisons on the RS effect removal against its nearest neighbor global shutter images.

### 4.2 Baseline Methods and Evaluation Metrics

**Baselines.** We compare our method against learning-free method DiffSfM (Zhuang et al., 2017) and several learning-based state-of-the-art rolling shutter effect removal methods, e.g. DSUN (Liu et al., 2020), SUNet (Fan et al., 2021), RSSR (Fan & Dai, 2021), CVR (Fan et al., 2022). Those learning-based methods usually take two consecutive images as input and train a deep network to recover the corresponding global shutter image. The network training usually requires a large dataset, which would be expensive/difficult to obtain in real scenarios. For fair comparisons, we use the official pre-trained models (of those baseline methods) for evaluations with Carla-RS dataset. However, for the newly synthesized datasets (e.g. Unreal-RS) as well as the real TUM-RS dataset, we are unable to

---

[†]We find that COLMAP (Schonberger & Frahm, 2016) is hardly to recover the poses of the Fastec-RS dataset Liu et al. (2020), which is used to initialize our method. We did not evaluate on this dataset.

Table 1: **Ablation studies for motion trajectory parameterization. USB-NeRF-lin-nodep** denotes the trajectory is parameterized with linear interpolation and there is no dependency between neighboring frames. **USB-NeRF-cub-nodep** denotes the trajectory is parameterized with cubic B-Spline and there is no dependency between neighboring frames. The experimental results demonstrate that cubic B-Spline parameterization performs better than linear interpolation in general, and the pose dependency between frames is also necessary.

|  | Carla | | | Blue Room | | | Living Room | | | White Room | | |
|---|---|---|---|---|---|---|---|---|---|---|---|---|
|  | PSNR↑ | SSIM↑ | LPIPS↓ | PSNR↑ | SSIM↑ | LPIPS↓ | PSNR↑ | SSIM↑ | LPIPS↓ | PSNR↑ | SSIM↑ | LPIPS↓ |
| USB-NeRF-lin-nodep | 16.78 | 0.551 | 0.2864 | 19.16 | 0.590 | 0.1546 | 16.42 | 0.580 | 0.3598 | 15.71 | 0.469 | 0.3327 |
| USB-NeRF-cub-nodep | 17.52 | 0.569 | 0.2501 | 17.77 | 0.544 | 0.2005 | 16.73 | 0.582 | 0.3375 | 15.89 | 0.471 | 0.2976 |
| USB-NeRF-linear | **32.15** | **0.892** | 0.0704 | 27.74 | 0.847 | 0.0928 | 29.01 | 0.858 | 0.1080 | 26.18 | 0.806 | 0.1040 |
| USB-NeRF-cubic | 31.90 | 0.889 | **0.0701** | **31.85** | **0.909** | **0.0573** | **34.89** | **0.939** | **0.0415** | **30.57** | **0.892** | **0.0576** |

fine-tune those methods due to a limited number of images. We, therefore, use the official pre-trained models for evaluations. Additionally, we also compare against the performance of the original NeRF (Mildenhall et al., 2020) and BARF (Lin et al., 2021) assuming the inputs are global shutter images with poses computed by COLMAP (Schonberger & Frahm, 2016).

**Evaluation metrics.** We evaluate the performance regarding rolling shutter effect removal and novel view image synthesis with the commonly used metrics, e.g. PSNR, SSIM and LPIPS (Zhang et al., 2018), between the recovered global shutter images and the ground truth global shutter images. We also compute the absolute trajectory error (ATE) against that estimated by COLMAP (Schonberger & Frahm, 2016), BARF (Lin et al., 2021) RSBA (Hedborg et al., 2012) and NW-RSBA (Liao et al., 2023), which is the most related to ours. The ATE error metric is commonly used for trajectory estimation evaluations in the visual odometry community.

## 4.3 Experimental Results

**Ablation studies** We evaluate four different camera pose interpolation strategies, to justify the advantage to use cubic B-Spline for whole trajectory parameterization. Besides cubic B-Spline, we also explore the linear interpolation strategy used in Hedborg et al. (2012), which assigns the first row of each RS image a pose parameter (i.e. $\mathbf{T}_i$) and then do linear interpolations for subsequent rows by using two neighboring poses (i.e. $\mathbf{T}_i$ and $\mathbf{T}_{i+1}$). These two strategies bring in additional constraints between neighboring frames, i.e. the pose of a particular row in the $i^{th}$ frame would also depends on the pose of the $(i + 1)^{th}$ frame. To relax this constraint, we also experiment with two additional strategies. We parameterize the camera trajectory for each RS image independently, i.e. assigning four control knots for each RS image for cubic B-Spline parameterization, and two control poses (i.e. $\mathbf{T}_{start}$ and $\mathbf{T}_{end}$) for the linear interpolation case.

We conduct experiments with the synthetic Carla-RS and the Unreal-RS datasets, by evaluating the rolling shutter effect removal performance in terms of PSNR, SSIM and LPIPS metrics. The experimental results are presented in Table 1. The results demonstrate that the optimization cannot be properly constrained if there is no pose dependency between neighboring frames. It can be explained by the degeneracy analysis for rolling shutter structure from motion (SfM) done by Albl et al. (2016), which states that near parallel readout directions of RS images would lead to degenerate solutions for SfM. The pose dependency by the other two parameterizations would bring in additional constraints to avoid the degenerate solutions, which are verified by the experimental results.

The results also reveal that cubic B-Spline interpolation performs similarly to linear interpolation if the camera moves at a constant velocity (e.g. Carla-RS dataset). However, it performs much better than linear interpolation, if the camera has realistic complex motions (e.g. Unreal-RS dataset). Therefore, we present experimental results for subsequent evaluations with the cubic B-Spline interpolation unless explicitly stated.

**Quantitative evaluation results.** We evaluate the performance of our method against other state-of-the-art methods in terms of rolling shutter effect removal and the accuracy of trajectory estimation, with both synthetic and real datasets. Table 2 presents the rolling shutter effect removal comparisons. It demonstrates that our method performs similarly to prior learning-based methods if those methods are trained on the respective dataset (e.g. Carla-RS dataset). However, those learning-based methods exhibit poor generalization performance when no fine-tuning on new dataset is performed (e.g. Unreal-RS dataset), while our method delivers good performance consistently, as our method does not rely on pre-training. Our method also performs better than learning-free method DiffSfM (Zhuang et al.,

Table 2: **Quantitative comparisons on the synthetic datasets in terms of rolling shutter effect removal.** Experimental results demonstrate that our method performs similarly to other learning-based methods on the Carla-RS dataset, on which those networks have been properly trained. However, our method performs much better on the Unreal-RS dataset, due to the poor generalization performance of other methods. USB-NeRF also performs better than the original NeRF and BARF, since they did not model the rolling shutter effect in their formulation.

| | Carla | | | Blue Room | | | Living Room | | | White Room | | |
|---|---|---|---|---|---|---|---|---|---|---|---|---|
| | PSNR↑ | SSIM↑ | LPIPS↓ | PSNR↑ | SSIM↑ | LPIPS↓ | PSNR↑ | SSIM↑ | LPIPS↓ | PSNR↑ | SSIM↑ | LPIPS↓ |
| DiffSfM(Zhuang et al., 2017) | 24.20 | 0.775 | 0.1322 | 17.10 | 0.497 | 0.2073 | 18.68 | 0.601 | 0.2236 | 14.94 | 0.469 | 0.2218 |
| DSUN (Liu et al., 2020) | 26.46 | 0.807 | 0.0703 | 21.25 | 0.682 | 0.1746 | 23.22 | 0.762 | 0.1507 | 20.45 | 0.643 | 0.1852 |
| SUNet (Fan et al., 2021) | 29.18 | 0.850 | 0.0658 | 23.32 | 0.721 | 0.1513 | 26.64 | 0.7957 | 0.1167 | 22.37 | 0.6866 | 0.1422 |
| RSSR (Fan & Dai, 2021) | 24.78 | 0.867 | 0.0695 | 22.15 | 0.731 | 0.1369 | 22.14 | 0.770 | 0.1258 | 19.19 | 0.697 | 0.1392 |
| CVR (Fan et al., 2022) | 31.74 | **0.929** | **0.0368** | 23.25 | 0.745 | 0.1268 | 23.14 | 0.785 | 0.1141 | 20.85 | 0.715 | 0.1212 |
| NeRF (Mildenhall et al., 2020) | 20.85 | 0.620 | 0.1734 | 19.12 | 0.569 | 0.3289 | 21.44 | 0.682 | 0.3495 | 18.29 | 0.509 | 0.4104 |
| BARF (Lin et al., 2021) | 20.95 | 0.664 | 0.1845 | 19.14 | 0.576 | 0.3446 | 21.48 | 0.690 | 0.3341 | 18.43 | 0.545 | 0.3837 |
| USB-NeRF (ours) | **31.90** | 0.889 | 0.0701 | **31.85** | **0.909** | **0.0573** | **34.89** | **0.939** | **0.0415** | **30.57** | **0.892** | **0.0576** |

Table 3: **Quantitative comparisons on both synthetic and real datasets in terms of the accuracy of trajectory estimation for translation error (m).** The experimental results demonstrate that rolling shutter distortions affect the accuracy of motion trajectory estimations. Due to proper modelling, our method performs much better than state-of-the-art methods. It also demonstrates that cubic B-Spline interpolation is superior to linear interpolation. The ATE metrics of the TUM-RS (Schubert et al., 2019) dataset are averaged over 10 sequences and details of every sequence are presented in the Appendix A.3 (Table 8). x denotes method failed on the corresponding sequence.

| | COLMAP | BARF | RSBA | NW-RSBA | USB-NeRF-linear | USB-NeRF-cubic |
|---|---|---|---|---|---|---|
| Carla | .1931±.1090 | .2245±.1293 | .1923±.0959 | .2720±.1404 | .0570±.0335 | **.0530±.0342** |
| Blue Room | .1593±.0949 | .1446±.0819 | .0640±.0308 | x | .0062±.0035 | **.0013±.0013** |
| Living Room | .0967±.0422 | .0919±.0400 | .1010±.0400 | x | .0144±.0089 | **.0035±.0025** |
| White Room | .1097±.0422 | .1191±.0521 | .1210±.0410 | x | .0115±.0067 | **.0044±.0033** |
| Adornment | .3269±.1952 | .3918±.2189 | x | x | .0536±.0834 | **.0162±.0122** |
| Factory | .2443±.1003 | .2149±.1326 | x | x | .0123±.0076 | **.0072±.0052** |
| Tanabata | .1397±.0745 | .1957±.1020 | x | x | .0154±.0085 | **.0130±.0077** |
| TUM-RS | .0486±.0228 | .0873±.0391 | .0688±.0322 | .1374±.0664 | .0150±.0104 | **.0136±.0090** |

2017) for all datasets. Additional experimental results in terms of rolling shutter effect removal on other synthetic sequences are presented in Appendix A.3 (Table 7).

The results shown in Table 2 also reveal that original NeRF (Mildenhall et al., 2020) and BARF (Lin et al., 2021) cannot deliver satisfying results if the rolling shutter image formation process is not explicitly considered. Even BARF (Lin et al., 2021) also optimizes the camera poses to eliminate the effect of inaccurate poses, it still cannot learn the correct underlying 3D representations, which delivers poor recovered global shutter images.

Table 3 presents the camera motion trajectory estimation results with both synthetic and real datasets. The results demonstrate that both COLMAP (Schonberger & Frahm, 2016) and BARF (Lin et al., 2021) suffer from the rolling shutter effect. The introduced distortions would affect camera pose estimations if they are not properly handled. On the contrary, our method does not have such limitations, since we formulate the physical image formation process of RS camera into the training of NeRF. Our method also performs better than SOTA RS-aware bundle adjustment methods (i.e. RSBA (Hedborg et al., 2012), NW-RSBA (Liao et al., 2023)) in terms of the ATE metric. The results also reveal that cubic B-Spline interpolation performs better than linear interpolation for both synthetic and real datasets, in terms of the accuracy of the recovered motion trajectories. Due to the limited space, we present detailed quantitative experimental results about trajectory estimation with real datasets in Appendix A.3 (Table 8).

More quantitative evaluation results (e.g. on novel view image synthesis, with un-ordered image sequences) are presented in Appendix A.3 (Table 5 and Table 6). They also demonstrate the better performance of our method over prior state-of-the-art methods.

**Qualitative evaluation results.** We also evaluate the qualitative performance of our method against the other baseline methods. Figure 4 presents the comparisons for both Carla-RS and Unreal-RS datasets. Figure 5 presents the results with the TUM-RS dataset (Schubert et al., 2019). Since the TUM-RS dataset does not provide pixel-aligned rolling-global shutter image pairs, we choose the nearest neighbor global shutter image (captured by another global shutter camera) for comparison.

The experimental results demonstrate that our method can better exploit multi-view information for rolling shutter effect removal, compared to DSUN (Liu et al., 2020), RSSR (Fan & Dai, 2021) and CVR (Fan et al., 2022) in Figure 4 (Carla-RS), even they are properly trained on the corresponding

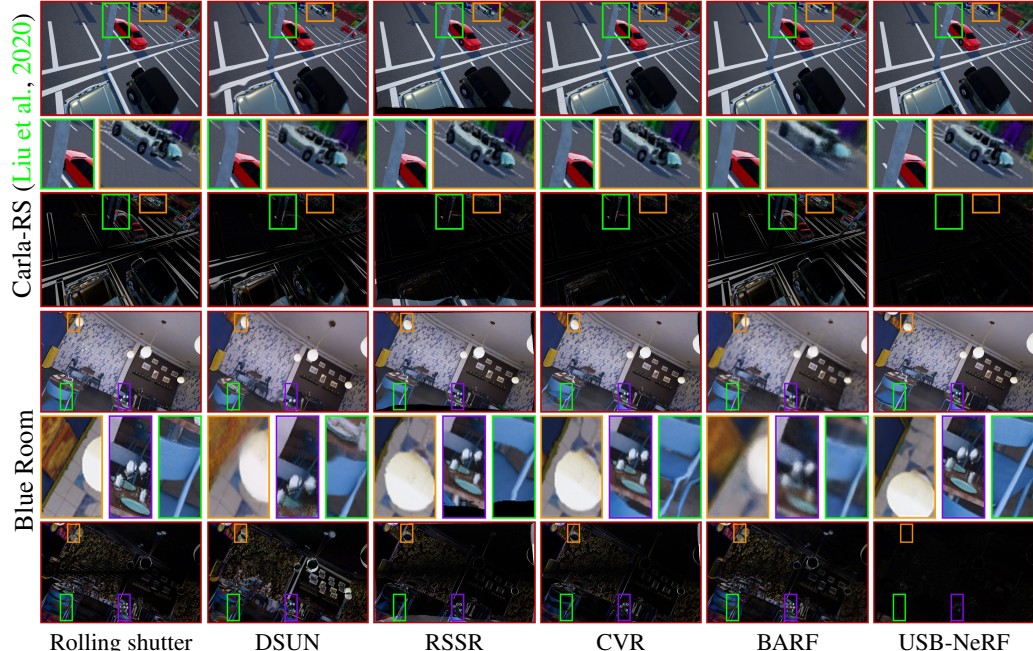

Figure 4: **Qualitative comparisons with Carla-RS datasets (Liu et al., 2020) and Unreal-RS datasets.** The experimental results demonstrate that our method achieves better performance compared to prior works. The darker the $3^{rd}$ and the $6^{th}$ rows, the performance is better.

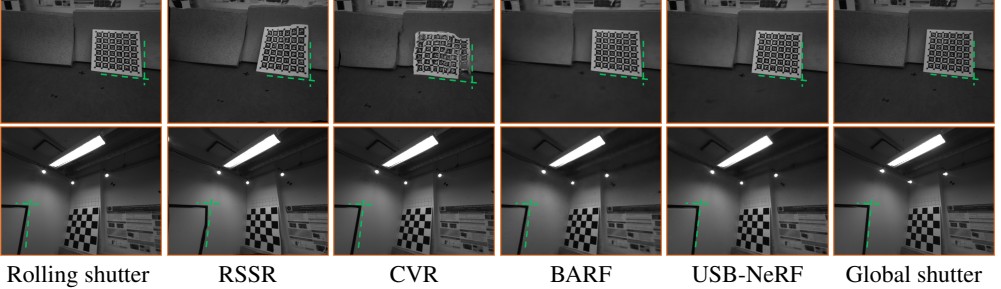

Figure 5: **Qualitative comparisons with real TUM-RS datasets (Schubert et al., 2019).** Since the dataset does not have pixel-aligned rolling-global shutter image pairs, we choose the nearest neighbor global shutter images for comparisons. The experimental results demonstrate that RSSR and CVR fail to correct the RS effect due to their poor generalization performance. BARF also fails since it does not consider the rolling shutter camera model, while our method successfully removes the RS effect.

dataset. Figure 4 (Blue Room) and Figure 5 demonstrate that both RSSR (Fan & Dai, 2021) and CVR (Fan et al., 2022) have a poor generalization performance if they are not fine-tuned on the respective datasets, which is common for practical applications. On the contrary, our method does not have such limitations and performs better than those learning-based RS effect removal methods consistently. The results also reveal that BARF (Lin et al., 2021) fails to learn the underlying undistorted 3D scene representation even though it optimizes the camera poses. It proves the necessity to properly model the physical image formation process of RS camera into the training of NeRF for better 3D scene reconstruction. More qualitative results can also be found in Appendix A.3 (e.g. novel view image synthesis, rolling shutter effect removal, trajectory estimation etc.) and supplementary video. They also demonstrate the superior performance of our method over prior works.

## 5 CONCLUSION

In this paper, we presented unrolling shutter bundle-adjusted neural radiance fields. The method takes advantage of the powerful representation ability of NeRF and a continuous-time trajectory representation with cubic B-Spline. Given a sequence of rolling shutter images, our method successfully learns the true underlying 3D representations and recovers the motion trajectory accurately. Experimental results demonstrate the superior performance of our method against prior state-of-the-art works, in terms of camera motion estimation, rolling shutter effect removal and novel view image synthesis etc.

ACKNOWLEDGMENTS

This work was supported in part by NSFC under Grant 62202389, in part by a grant from the Westlake University-Muyuan Joint Research Institute, and in part by the Westlake Education Foundation.

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

## A APPENDIX

In the appendix, we present the details on method implementation and training, frame selections from TUM-RS (Schubert et al., 2019) datasets, more experimental results on image quality of synthesized novel views and visualization of estimated motion trajectories. The rendered novel view high frame-rate global shutter video is presented in the supplementary video. We will present each part as follows.

### A.1 IMPLEMENTATION AND TRAINING DETAILS

We implement our method in PyTorch. We use Adam (Kingma & Ba, 2014) optimizer to estimate the weights of the MLP network and the pose parameters of the spline. The optimizer is configured with $\beta_1 = 0.9$ and $\beta_2 = 0.999$ for NeRF and pose optimizations. We set the learning rates to be $5 \times 10^{-4}$ and $1 \times 10^{-3}$ for the NeRF and pose optimizers respectively. Both learning rates gradually decay to $5 \times 10^{-5}$ and $1 \times 10^{-5}$ respectively. During each training step, we randomly select 7200 pixels from all training images to minimize the loss function presented in Eq. 10 and we run a total of 200K steps on an NVIDIA RTX 3090 GPU. We adopt the linear adjustment of the positional encoding starting from steps 20K to 100K to achieve the coarse-to-fine training strategy as in BARF (Lin et al., 2021). We select the pose corresponding to the first row of each image as the control knots of the spline, and they are initialized with the poses computed via COLMAP (Schonberger & Frahm, 2016) from the sequence of rolling shutter images.

### A.2 DETAILS ON FRAME SELECTIONS FROM TUM-RS DATASETS

Since the total number of frames of each TUM-RS (Schubert et al., 2019) sequence is too long to be processed by NeRF (Mildenhall et al., 2020) and our method, we choose sub-sequence frames to evaluate our method. The details are listed in Table 4.

Table 4: **Selected frames from each sequence of TUM-RS datasets (Schubert et al., 2019).**

| Seq. | Start frame timestamp | End frame timestamp |
|------|----------------------|---------------------|
| 1  | 1548685771426550686 | 1548685773576617686 |
| 2  | 1548685824354495627 | 1548685826504562627 |
| 3  | 1548685846189303526 | 1548685848339371526 |
| 4  | 1548685907331049323 | 1548685909481116323 |
| 5  | 1548686030960692141 | 1548686033110759141 |
| 6  | 1548689768046781352 | 1548689770196849352 |
| 7  | 1548689850274927799 | 1548689852424994799 |
| 8  | 1548689861503747264 | 1548689863653814264 |
| 9  | 1548689939497068095 | 1548689941647135095 |
| 10 | 1548689997870607830 | 1548690000020674830 |

Table 5: **Quantitative comparisons on the synthetic datasets in terms of novel view synthesis.** Experimental results demonstrate that our method is able to synthesize high-quality novel view global shutter images.

| | Blue Room | | | Living Room | | | White Room | | |
|---|---|---|---|---|---|---|---|---|---|
| | PSNR↑ | SSIM↑ | LPIPS↓ | PSNR↑ | SSIM↑ | LPIPS↓ | PSNR↑ | SSIM↑ | LPIPS↓ |
| NeRF+DiffSfM(Zhuang et al., 2017) | 13.87 | 0.420 | 0.7271 | 13.35 | 0.398 | 0.6415 | 14.11 | 0.467 | 0.5926 |
| NeRF+DSUN (Liu et al., 2020) | 18.08 | 0.550 | 0.4391 | 17.47 | 0.574 | 0.5005 | 18.96 | 0.592 | 0.3821 |
| NeRF+SUNet (Fan et al., 2021) | 19.52 | 0.641 | 0.3732 | 18.69 | 0.662 | 0.4031 | 14.87 | 0.412 | 0.5885 |
| NeRF+RSSR (Fan & Dai, 2021) | 17.86 | 0.566 | 0.3175 | 18.73 | 0.651 | 0.3628 | 11.83 | 0.296 | 0.6422 |
| NeRF+CVR (Fan et al., 2022) | 15.94 | 0.464 | 0.4968 | 15.88 | 0.503 | 0.5753 | 20.67 | 0.703 | 0.2506 |
| NeRF (Mildenhall et al., 2020) | 18.56 | 0.562 | 0.3820 | 15.62 | 0.446 | 0.6098 | 15.60 | 0.426 | 0.5616 |
| BARF (Lin et al., 2021) | 19.13 | 0.540 | 0.4592 | 16.03 | 0.544 | 0.5296 | 12.94 | 0.389 | 0.7600 |
| USB-NeRF (ours) | **28.99** | **0.886** | **0.0757** | **33.33** | **0.936** | **0.0468** | **29.11** | **0.889** | **0.0598** |

## A.3 ADDITIONAL EXPERIMENTAL RESULTS

To evaluate the performance of our method for novel view image synthesis, we select 6 additional views from each scene of Unreal-RS datasets. Since DiffSfM (Zhuang et al., 2017), DSUN (Liu et al., 2020), SUNet (Fan et al., 2021), RSSR (Fan & Dai, 2021) and CVR(Fan et al., 2022) cannot synthesize novel view image, we first apply these approaches to restore global shutter images, and then train the original NeRF (Mildenhall et al., 2020). Table 5 and Figure 6 demonstrate that our method outperforms prior methods in terms of novel view image synthesis. The poor performances of DSUN (Liu et al., 2020), SUNet (Fan et al., 2021), RSSR (Fan & Dai, 2021) and CVR(Fan et al., 2022), are caused by their poor generalization capabilities on domain-shifted datasets. Learning-free method DiffSfM (Zhuang et al., 2017) similarly shows poor performance on all datasets as it could not realize perfect rolling shutter effect removal even with bundle adjustment. The experimental results also reveal that both NeRF (Mildenhall et al., 2020) and BARF (Lin et al., 2021) cannot perform well either, due to their ignorance of the rolling shutter effect presented in the training images. Since the Carla-RS (Liu et al., 2020) does not provide additional images for novel view image synthesis evaluation, we only present the qualitative results in Figure 7. It also demonstrates that our method could take advantage of multi-view information instead of only two views, and thus performs better than existing methods in terms of novel view image synthesis.

To further evaluate our method, we also train USB-NeRF with un-ordered images (i.e. there is no pose dependency among input frames). As mentioned in our main paper and proved by Albl et al. (2016), rolling shutter bundle adjustment with un-ordered images would have degenerated solutions if

Table 6: **Quantitative comparisons on the unorganized synthetic datasets.** The experimental results demonstrate that our method also performs better than prior methods with un-ordered rolling shutter images, in terms of rolling shutter effect correction.

| | Blue Room | | | Living Room | | | Roof | | |
|---|---|---|---|---|---|---|---|---|---|
| | PSNR↑ | SSIM↑ | LPIPS↓ | PSNR↑ | SSIM↑ | LPIPS↓ | PSNR↑ | SSIM↑ | LPIPS↓ |
| NeRF (Mildenhall et al., 2020) | 20.10 | 0.601 | 0.3177 | 24.98 | 0.784 | 0.2364 | 19.10 | 0.489 | 0.5200 |
| BARF (Lin et al., 2021) | 20.28 | 0.591 | 0.3608 | 24.58 | 0.776 | 0.2533 | 19.14 | 0.493 | 0.5234 |
| USB-NeRF | **31.13** | **0.900** | **0.0615** | **32.46** | **0.923** | **0.0364** | **27.26** | **0.747** | **0.1469** |

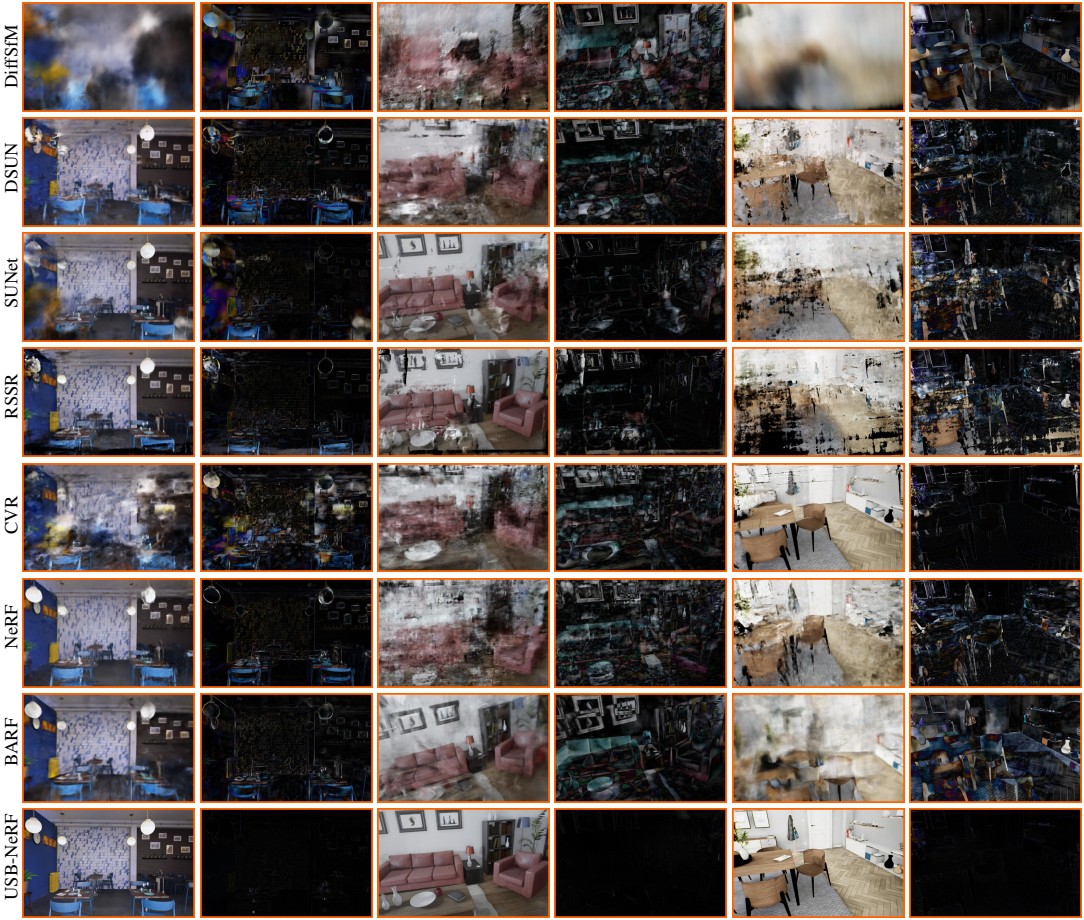

Figure 6: **Qualitative comparisons with synthetic datasets for novel view synthesis.** The experimental results demonstrate the ability of our approach to synthesize novel view images in good quality. The even columns present the error residual images between the rendered and ground-truth global shutter images. The darker the better.

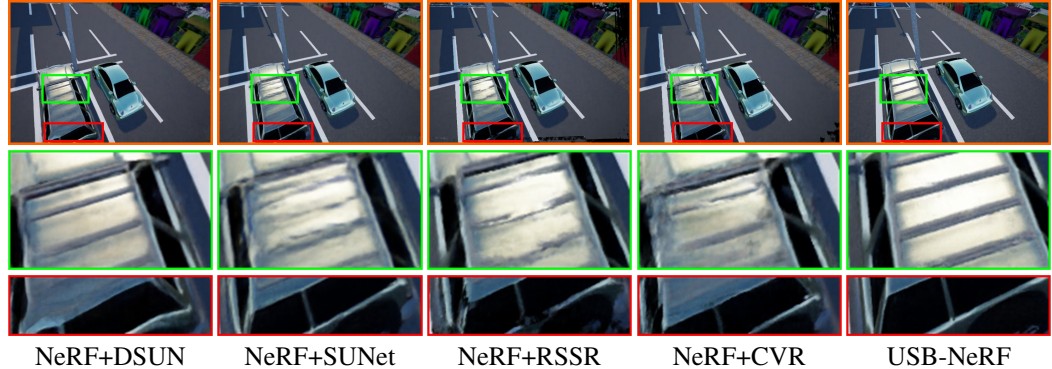

Figure 7: **Qualitative comparisons with Carla-RS datasets for novel view synthesis.** The experimental results demonstrate the ability of our approach to synthesize novel view images in good quality, by taking advantage of multi-view images.

the input images are not properly captured. Thus, we follow the instructions suggested by Albl et al. (2016), i.e. the mutual angle between readout directions should be larger than 30 degrees, to generate the training dataset. We synthesized 3 sequences of un-ordered rolling shutter images with Unreal game engine (i.e. Unreal-RS-BlueRoom, Unreal-RS-LivingRoom, Unreal-RS-Roof) in total. During

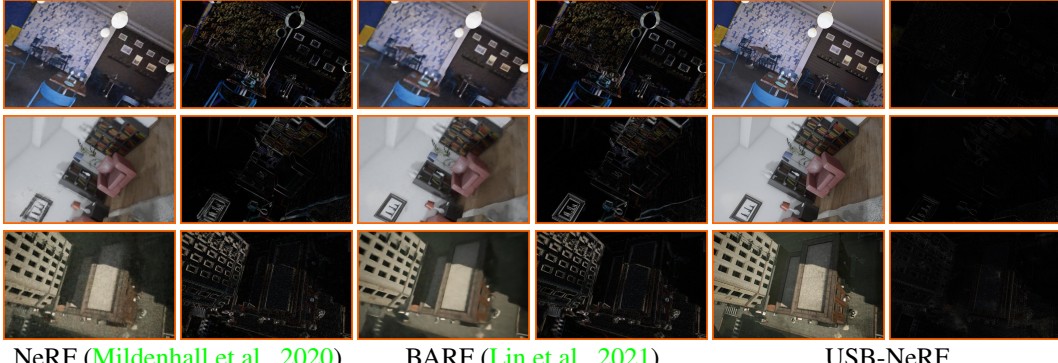

NeRF (Mildenhall et al., 2020)  BARF (Lin et al., 2021)  USB-NeRF

Figure 8: **Qualitative comparisons with unordered Unreal-RS datasets.** The experimental results demonstrate that our approach could also generate high-quality global shutter images with a set of un-ordered rolling shutter images. The even columns present the error residual images between the rendered and ground-truth global shutter images. The darker the better.

Table 7: **Quantitative comparisons on the synthetic datasets in terms of rolling shutter effect removal.** Experimental results demonstrate that our method also achieves better performance compared to previous methods on these additional sequences.

| | Adornment | | | Factory | | | Tanabata | | |
|---|---|---|---|---|---|---|---|---|---|
| | PSNR↑ | SSIM↑ | LPIPS↓ | PSNR↑ | SSIM↑ | LPIPS↓ | PSNR↑ | SSIM↑ | LPIPS↓ |
| DiffSfM(Zhuang et al., 2017) | 14.91 | 0.395 | 0.3012 | 15.21 | 0.406 | 0.2310 | 11.63 | 0.337 | 0.3174 |
| DSUN(Liu et al., 2020) | 17.84 | 0.534 | 0.2603 | 19.75 | 0.565 | 0.1835 | 16.06 | 0.493 | 0.2135 |
| SUNet(Fan et al., 2021) | 19.11 | 0.563 | 0.2273 | 22.84 | 0.647 | 0.1411 | 17.79 | 0.558 | 0.1832 |
| RSSR(Fan & Dai, 2021) | 18.56 | 0.602 | 0.1892 | 19.34 | 0.631 | 0.1438 | 16.06 | 0.566 | 0.1660 |
| CVR(Fan et al., 2022) | 19.70 | 0.620 | 0.1774 | 20.89 | 0.649 | 0.1292 | 17.68 | 0.581 | 0.1480 |
| NeRF (Mildenhall et al., 2020) | 17.28 | 0.473 | 0.6193 | 18.02 | 0.459 | 0.4621 | 15.04 | 0.369 | 0.5828 |
| BARF (Lin et al., 2021) | 17.46 | 0.485 | 0.5614 | 17.94 | 0.459 | 0.436 | 15.28 | 0.351 | 0.6742 |
| USB-NeRF (ours) | **29.97** | **0.876** | **0.0892** | **32.67** | **0.898** | **0.0819** | **23.84** | **0.750** | **0.1947** |

experiments, we represent the camera motion within individual frame readout time by cubic B-Spline with 4 control knots. Table 6 and Figure 8 present the experimental results. Since prior rolling shutter effect removal networks usually rely on two consecutive frames to restore the global shutter image, they are not suitable for this dataset. We only evaluate our method against both NeRF (Mildenhall et al., 2020) and BARF (Lin et al., 2021). The experimental results demonstrate that our method also outperforms prior methods with unordered rolling shutter images.

Table 8: **Quantitative comparisons on TUM-RS datasets (Schubert et al., 2019) in terms of the Absolute Trajectory Error (ATE) metric (m).** The experimental results demonstrate that our method performs much better than prior works in terms of the accuracy of motion trajectory estimation.

| | COLMAP | BARF | RSBA | NW-RSBA | USB-NeRF-linear | USB-NeRF-cubic |
|---|---|---|---|---|---|---|
| seq-1 | .0237±.0233 | .0345±.0176 | .0136±.0074 | .1162±.0340 | .0047±.0029 | **.0038**±.0021 |
| seq-2 | .1432±.0676 | .1847±.0635 | .3574±.1713 | .4349±.2047 | .0592±.0506 | **.0560**±.0332 |
| seq-3 | .0476±.0340 | .0743±.0437 | **.0106**±.0057 | .0123±.0065 | .0120±.0060 | .0111±.0060 |
| seq-4 | .0180±.0059 | .0294±.0113 | .0064±.0026 | .0473±.0289 | .0064±.0026 | **.0050**±.0023 |
| seq-5 | .0662±.0275 | .0999±.0394 | .0144±.0048 | .0366±.0554 | .0119±.0043 | **.0114**±.0041 |
| seq-6 | .0349±.0124 | .0719±.0451 | .0137±.0078 | .0626±.0327 | .0163±.0074 | **.0035**±.0021 |
| seq-7 | .0184±.0065 | .0185±.0057 | .0056±.0034 | .0700±.0289 | .0036±.0015 | **.0030**±.0012 |
| seq-8 | .0417±.0189 | .0638±.0340 | .0102±.0040 | .1787±.0863 | .0096±.0069 | **.0095**±.0070 |
| seq-9 | .0512±.0191 | .1509±.0835 | .2432±.1061 | .2780±.1186 | **.0128**±.0130 | .0150±.0166 |
| seq-10 | .0417±.0122 | .1450±.0475 | **.0126**±.0090 | .1378±.0682 | .0135±.0085 | .0178±.0149 |
| Avg. | .0486±.0228 | .0873±.0391 | .0688±.0322 | .1374±.0664 | .0150±.0104 | **.0136**±.0090 |

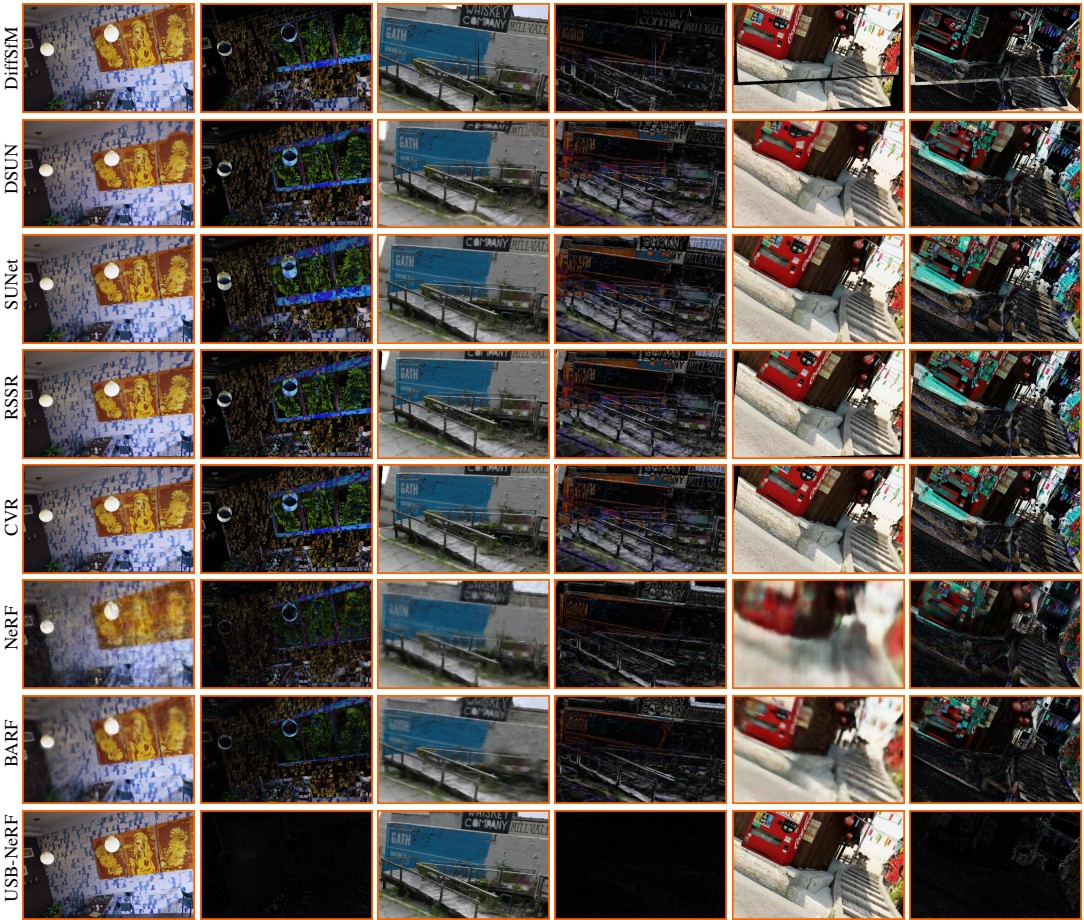

Figure 9: **Qualitative comparisons on synthetic datasets for rolling Shutter effect removal.** The experimental results demonstrate that our method is superior than all prior works. The even columns present the error residual images between the rendered and ground-truth global shutter images. The darker the better.

Table 9: **Quantitative comparisons on synthetic datasets in terms of the Relative Pose Error for rotation part (°/frame).** The experimental results demonstrate that rolling shutter distortions affect the accuracy of motion trajectory estimations. Due to proper modeling, our method performs much better than state-of-the-art methods. It also demonstrates that cubic B-Spline interpolation is superior to linear interpolation. x denotes method failed on the corresponding sequence.

|  | COLMAP | BARF | RSBA | NW-RSBA | USB-NeRF-linear | USB-NeRF-cubic |
|---|---|---|---|---|---|---|
| Carla | $2.357 \pm 1.647$ | $2.762 \pm 2.041$ | $1.844 \pm 1.080$ | $2.090 \pm 1.492$ | $0.586 \pm 0.174$ | $\mathbf{0.528 \pm 0.157}$ |
| BlueRoom | $2.594 \pm 1.411$ | $8.123 \pm 3.585$ | $7.257 \pm 3.413$ | x | $0.127 \pm 0.074$ | $\mathbf{0.046 \pm 0.041}$ |
| LivingRoom | $2.145 \pm 1.320$ | $9.016 \pm 6.243$ | $7.405 \pm 4.383$ | x | $0.381 \pm 0.140$ | $\mathbf{0.108 \pm 0.053}$ |
| WhiteRoom | $1.810 \pm 0.771$ | $4.702 \pm 2.077$ | $4.212 \pm 1.631$ | x | $0.194 \pm 0.127$ | $\mathbf{0.089 \pm 0.039}$ |
| Adornment | $2.723 \pm 1.438$ | $10.390 \pm 4.659$ | x | x | $0.581 \pm 1.273$ | $\mathbf{0.234 \pm 0.147}$ |
| Factory | $2.194 \pm 1.187$ | $8.523 \pm 4.028$ | x | x | $0.209 \pm 0.112$ | $\mathbf{0.135 \pm 0.080}$ |
| Tanabata | $2.744 \pm 1.662$ | $15.187 \pm 13.197$ | x | x | $1.042 \pm 0.459$ | $\mathbf{1.008 \pm 0.463}$ |
| Avg. | $2.367 \pm 1.348$ | $8.386 \pm 5.119$ | $5.179 \pm 2.627$ | $2.090 \pm 1.492$ | $0.446 \pm 0.337$ | $\mathbf{0.307 \pm 0.140}$ |

Additional experimental results on more synthetic datasets in terms of the rolling shutter effect removal are also presented in Table 7 and Figure 9. It demonstrates that our method also performs better than prior state-of-the-art methods. Additional results on real-world datasets captured using GoPro HERO6 Black, Canon camera (EOS M3), and iPhone 14 Pro are presented in Figure 10, Figure 11 and Figure 12. The results on real-world dataset demonstrate that both NeRF(Mildenhall et al., 2020) and BARF(Lin et al., 2021) fail to correct the RS distortion, while our method renders correct global shutter images with no artifact.

Table 10: **Quantitative comparisons on real datasets in terms of the Relative Pose Error for rotation part (°/frame).** The experimental results demonstrate that rolling shutter distortions affect the accuracy of motion trajectory estimations. Due to proper modeling, our method performs much better than state-of-the-art methods. It also demonstrates that cubic B-Spline interpolation is superior to linear interpolation. x denotes method failed on the corresponding sequence.

| | COLMAP | BARF | RSBA | NW-RSBA | USB-NeRF-linear | USB-NeRF-cubic |
|---|---|---|---|---|---|---|
| seq1 | $0.278 \pm 0.284$ | $0.283 \pm 0.152$ | $0.578 \pm 0.470$ | $0.671 \pm 0.545$ | $0.114 \pm 0.072$ | $\mathbf{0.108 \pm 0.071}$ |
| seq2 | $0.794 \pm 0.428$ | $\mathbf{0.737 \pm 0.432}$ | $9.789 \pm 14.344$ | $28.325 \pm 40.662$ | $1.365 \pm 1.733$ | $0.777 \pm 0.838$ |
| seq3 | $0.271 \pm 0.145$ | $0.272 \pm 0.176$ | $0.397 \pm 0.208$ | $0.250 \pm 0.116$ | $0.159 \pm 0.115$ | $\mathbf{0.145 \pm 0.108}$ |
| seq4 | $0.225 \pm 0.100$ | $0.211 \pm 0.110$ | $0.247 \pm 0.126$ | $1.049 \pm 0.571$ | $0.160 \pm 0.093$ | $\mathbf{0.155 \pm 0.093}$ |
| seq5 | $0.607 \pm 0.286$ | $0.545 \pm 0.199$ | $2.245 \pm 1.784$ | $3.131 \pm 2.316$ | $0.168 \pm 0.087$ | $\mathbf{0.166 \pm 0.082}$ |
| seq6 | $0.345 \pm 0.163$ | $0.322 \pm 0.247$ | $0.575 \pm 0.257$ | $0.851 \pm 0.874$ | $0.180 \pm 0.089$ | $\mathbf{0.149 \pm 0.080}$ |
| seq7 | $0.219 \pm 0.123$ | $0.224 \pm 0.143$ | $0.324 \pm 0.201$ | $1.063 \pm 0.700$ | $0.146 \pm 0.100$ | $\mathbf{0.128 \pm 0.097}$ |
| seq8 | $0.520 \pm 0.325$ | $0.249 \pm 0.103$ | $0.502 \pm 0.251$ | $1.718 \pm 1.011$ | $0.151 \pm 0.069$ | $\mathbf{0.135 \pm 0.066}$ |
| seq9 | $0.322 \pm 0.165$ | $0.445 \pm 0.276$ | $4.400 \pm 7.312$ | $41.983 \pm 47.732$ | $\mathbf{0.193 \pm 0.211}$ | $0.263 \pm 0.524$ |
| seq10 | $0.351 \pm 0.165$ | $1.006 \pm 2.460$ | $1.889 \pm 1.117$ | $4.752 \pm 4.748$ | $\mathbf{0.193 \pm 0.277}$ | $0.250 \pm 0.499$ |
| Avg. | $0.393 \pm 0.218$ | $0.429 \pm 0.430$ | $2.095 \pm 2.607$ | $8.379 \pm 9.928$ | $0.283 \pm 0.285$ | $\mathbf{0.227 \pm 0.246}$ |

Table 8 presents the details on trajectory estimation in terms of the ATE metric for translation error with the TUM-RS dataset. Table 9 and Table 10 presents the details of RPE metric for rotation error. The results show that our method performs consistently better than both COLMAP and BARF. It also demonstrates that our method is able to perform on-par against prior rolling shutter bundle adjustment methods, i.e. RSBA and NW-RSBA, and achieves better performance in terms of average ATE and RPE metrics over all sequences.

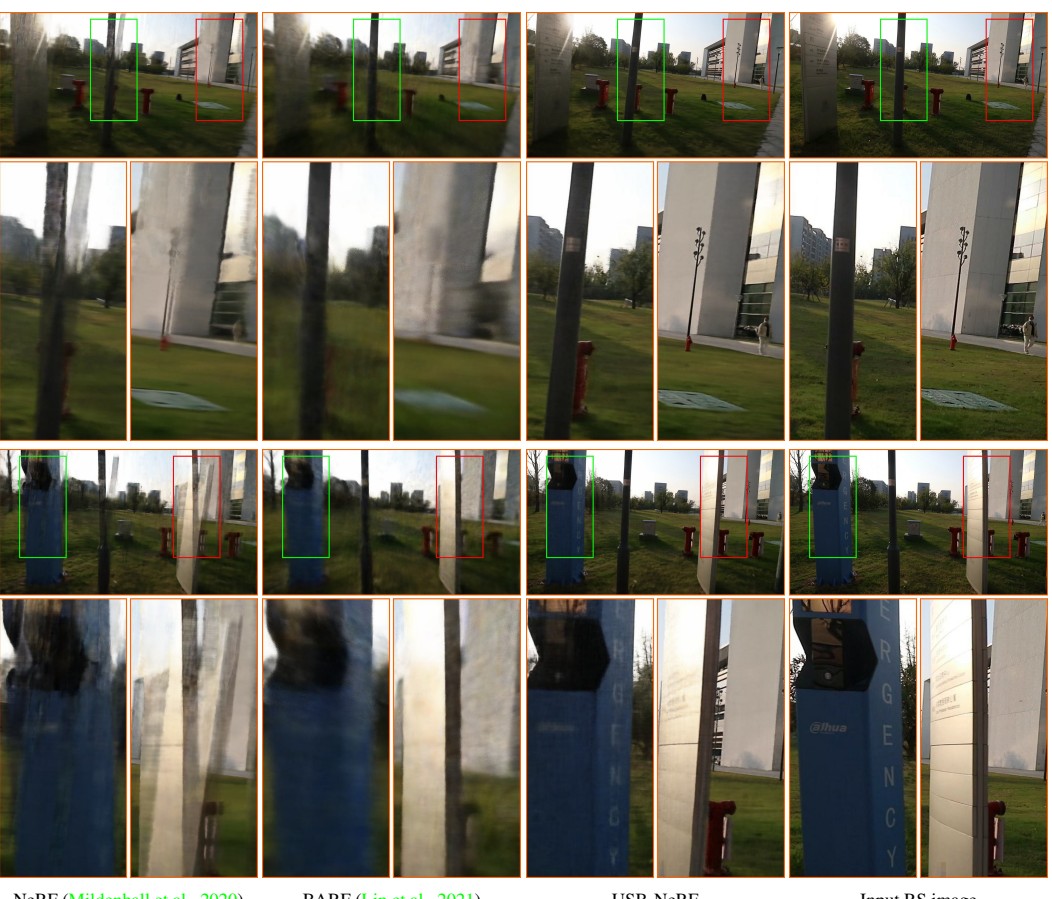

NeRF (Mildenhall et al., 2020)  BARF (Lin et al., 2021)  USB-NeRF  Input RS image

Figure 10: **Qualitative comparisons on real-world datasets captured by a Canon camera.** The camera exhibits forward and backward motion, which challenges both NeRF and BARF to recover the true underlying 3D scene representation. Our method recovers the correct global shutter images. Note the camera is not parallel to the ground during capture.

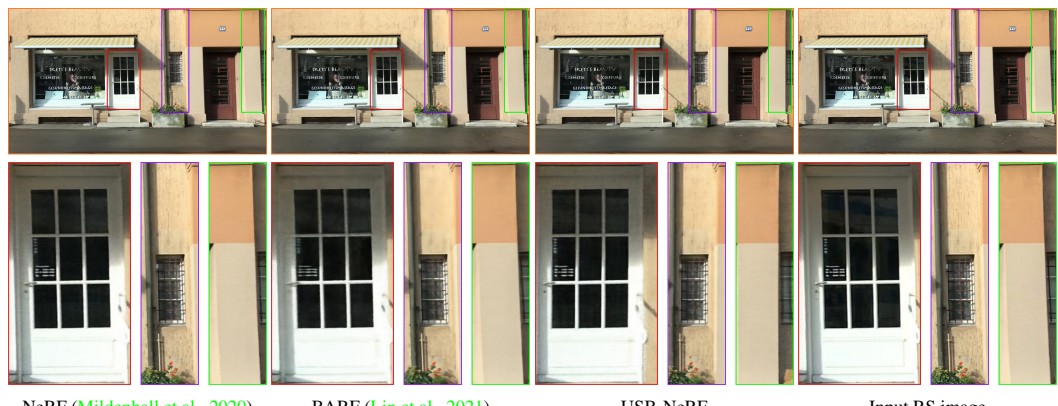

NeRF (Mildenhall et al., 2020)    BARF (Lin et al., 2021)    USB-NeRF    Input RS image

Figure 11: **Qualitative comparisons on real-world datasets captured by GoPro HERO6.** The images are captured on a moving tram in constant direction. It demonstrates that even both NeRF and BARF can render clear images, the recovered scene is distorted. Our method can correctly un-distort it.

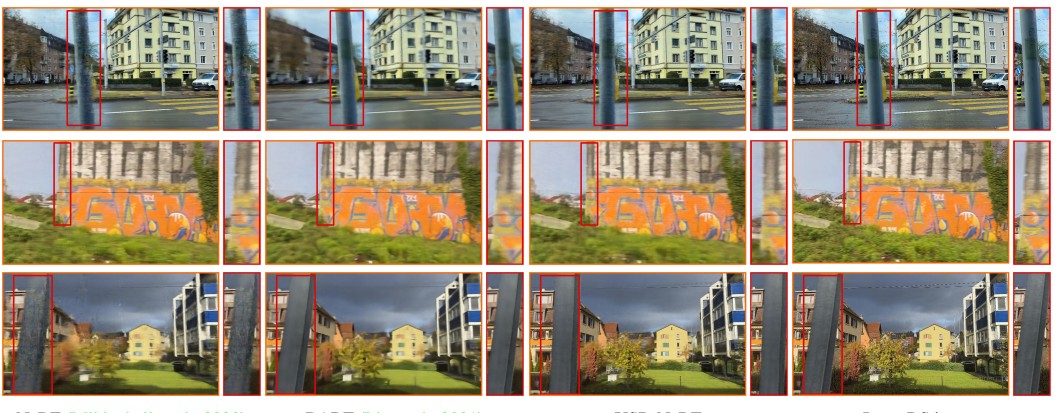

NeRF (Mildenhall et al., 2020)    BARF (Lin et al., 2021)    USB-NeRF    Input RS image

Figure 12: **Qualitative comparisons on real-world datasets captured by iPhone 14 Pro.** The images are captured on a moving bus in constant direction. It demonstrates that both NeRF and BARF fail to correct the distortion and produce additional artifacts, while our method successfully restores the true underlying 3D scene representation.

## A.4 TRAJECTORY VISUALIZATION

We also present additional qualitative results in terms of motion trajectory estimations. The experiments are conducted via the real TUM-RS datasets (Schubert et al., 2019). The experimental results shown in Figure 13 and Figure 14 demonstrate that USB-NeRF is able to recover the motion trajectories on TUM-RS (Schubert et al., 2019) datasets.

## A.5 HIGH-FRAME GLOBAL SHUTTER VIDEOS

To further demonstrate the advantage of our method, we also present a supplementary video which demonstrates the ability of our method to recover high quality high frame-rate global shutter images from a single rolling shutter image, which encodes rich temporal information. The video is attached as a separate file. The results also demonstrate the superior performance of our method against prior state-of-the-art methods.

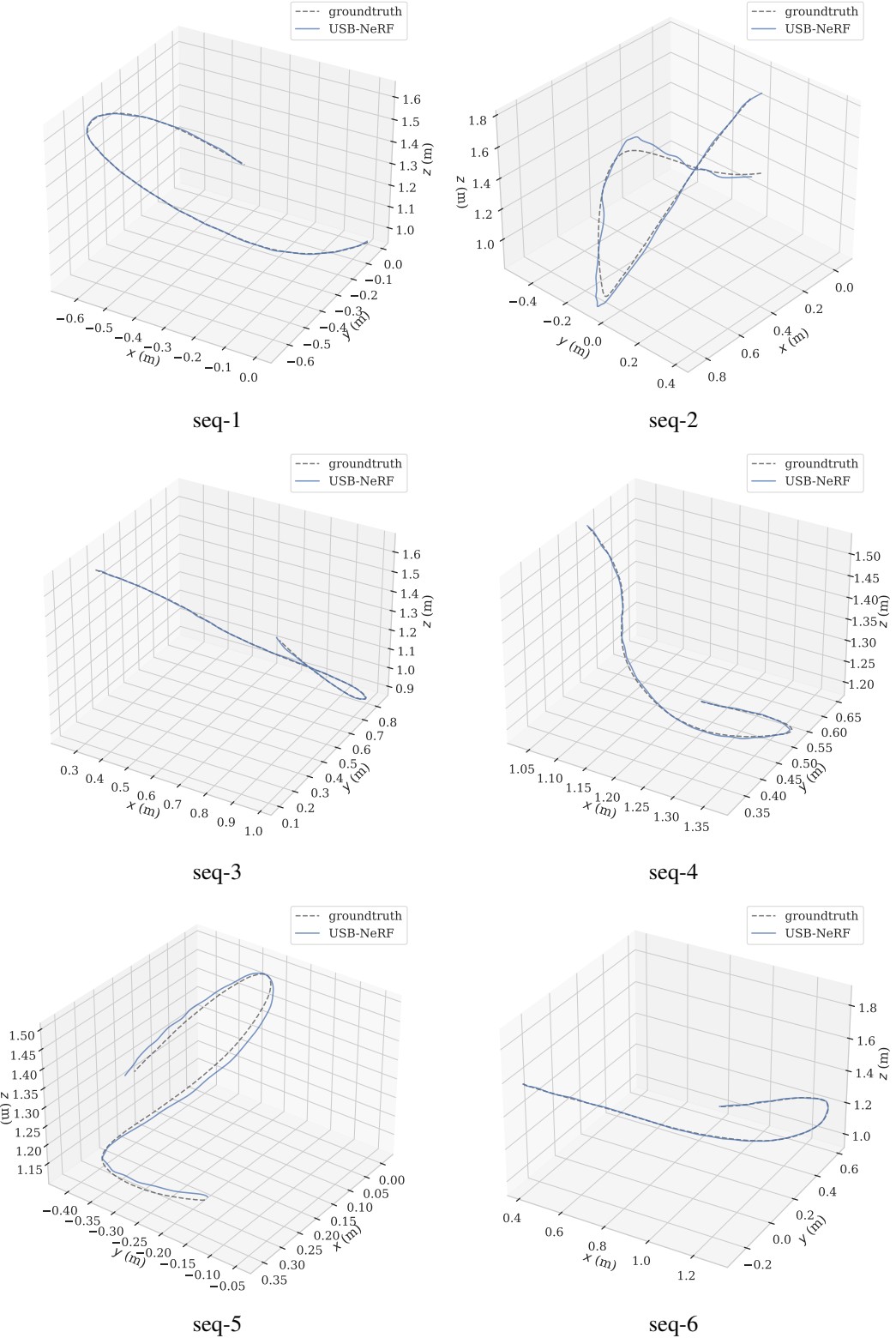

Figure 13: **Comparisons of estimated trajectories for sequence 1-6 of real TUM-RS datasets (Schubert et al., 2019).** The experimental results demonstrate that our method is able to estimate the motion trajectories with a sequence of rolling shutter images.

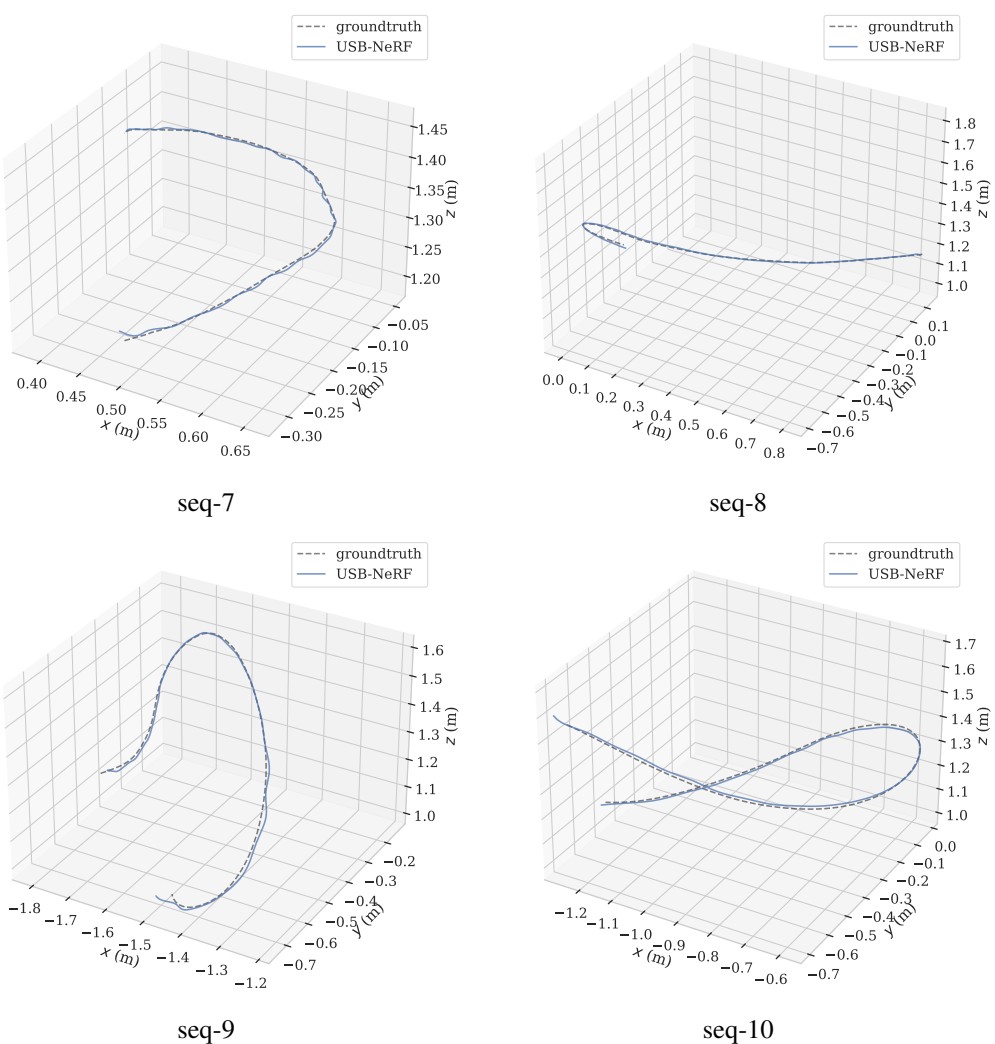

Figure 14: **Comparisons of estimated trajectories for sequence 7-10 of real TUM-RS datasets (Schubert et al., 2019).** The experimental results demonstrate that our method is able to estimate the motion trajectories with a sequence of rolling shutter images.

