# OpenReview forum: "USB-NeRF: Unrolling Shutter Bundle Adjusted Neural Radiance Fields"
_ICLR.cc/2024/Conference — ICLR 2024 poster_

### Official Review · Reviewer_Xdqv · 2023-10-28

**Soundness:** 2 fair
**Presentation:** 3 good
**Contribution:** 1 poor
**Rating:** 3
**Confidence:** 3

**Summary:**

# Summary
This paper integrates the rolling shutter model into NeRF training. The proposed method assumes the given images are rolling shutter images. Given the specific camera motion trajectory model, the rendered rolling shutter images can be computed from the global shutter images from the NeRF model. The camera trajectory and the NeRF model can be optimized by minimizing the loss between rendered rolling shutter images and the input images.

**Strengths:**

# Strength
- good performance. Since the experiments are done on the rolling shutter dataset, it is unsurprising that the proposed method outperforms previous methods like barf.
- The idea is straightforward, like some previous NeRF-based methods integrating another luminance/render/texture/deblur model, like BRDF.

**Weaknesses:**

# Weakness
- The contribution and the novelty are limited. The effect of the rolling shutter is well-known in 3D vision community, and many works have been trying to solve it in the past years. The proposed methods only contain the basic concept of modeling a rolling shutter.
- It seems like the proposed method uses the specific motion model of the camera. It might prevent the proposed method from working on the global shutter dataset(no experiment to prove it) and another dataset that cannot be modeled in bicubic motion(no experiment to prove it).
- The proposed method requires COLMAP to provide an initial camera pose, which is only mentioned in the footnote.
- To sum up, due to the abovementioned concerns, I cannot give a positive rating to the proposed method in the current version.

**Questions:**

A Possible Direction for improvement is to regard the camera motion model as the parameters in the Nerf model and solve it during optimization. To this end, the proposed method can be more general to handle global and rolling shutter datasets. It will be a benefit for the community further.

---

> ### Author Response · Authors · 2023-11-20
>
> We thank the reviewer for the acknowledgement of the good performance of our method and the effort spent in reviewing our paper. We make replies to the questions from the reviewer as follows:
>
>
>  **Q1: The contribution and the novelty are limited. The effect of the rolling shutter is well-known in 3D vision community, and many works have been trying to solve it in the past years. The proposed methods only contain the basic concept of modeling a rolling shutter.**
>
> A1: Although rolling shutter effect correction has been researched for many years, prior methods still face challenges to solve it. In contrary, we are the first to formulate the problem under the framework of NeRF and deliver very promising results for the problem.
>
>
>  **Q2: It seems like the proposed method uses the specific motion model of the camera. It might prevent the proposed method from working on the global shutter dataset(no experiment to prove it) and another dataset that cannot be modeled in bicubic motion(no experiment to prove it).**
>
> A2-1: We conduct experiments with global shutter dataset to prove that our formulation could also work for global shutter images. We exploit three public datasets for NeRF with global shutter images. The results are presented in the following table. It demonstrates that our formulation has comparable performance as prior state-of-the-art methods on global shutter dataset. Note that those methods cannot work on RS dataset as ours.
>
> | |  |  fern  | |  |  flower |  |  |  fortress    |  | |  Avg.  |  |
> |:--:|:--:|:--:|:--:|:--:|:--:|:--:|:--:|:--:|:--:|:--:|:--:|:-:|
> | | PSNR$\uparrow$ | SSIM$\uparrow$ | LPIPS$\downarrow$ | PSNR$\uparrow$ | SSIM$\uparrow$ | LPIPS$\downarrow$ | PSNR$\uparrow$ | SSIM$\uparrow$ | LPIPS$\downarrow$ | PSNR$\uparrow$ | SSIM$\uparrow$ | LPIPS$\downarrow$ |
> | NeRF |  26.12 |  0.825 | **0.0833**    |    **30.79**   |    **0.904**   | **0.0218**    |  22.29 |    **0.836**   | **0.0791**    |    **26.40**   |    **0.855**   | **0.0614**    |
> | BARF    |  25.24 |  0.775 |   0.1953  |  29.54 |  0.846 |   0.0609  |    **22.96**   |  0.740 |   0.1732  |  25.91 |  0.787 |   0.1431  |
> |  USB-NeRF |    **26.31**   |    **0.830**   |   0.0863  |  29.51 |  0.836 |   0.0625  |  22.57 |  0.813 |   0.1051  |  26.13 |  0.827 |   0.0846  |
>
> A2-2: We agree that the cubic B-Spline motion model would not work well for certain very specific motions, such as highly non-linear motions. However, we think cubic B-Spline could handle most real world application scenarios, in which the camera motions are usually smooth.
>
>
>  **Q3: The proposed method requires COLMAP to provide an initial camera pose, which is only mentioned in the footnote.**
>
> A3: We indeed also mentioned it in the ``Implementation and training details'' of the paper. Due to space limit, we put it into our Appendix, i.e. A1, as most prior papers did. Furthermore, using COLMAP to compute an initial pose estimation would make the training more stable, which is also a common practice by most prior works. We do not think it would be a limitation. For certain scenarios, we could also relax the requirement of COLMAP, e.g. to integrate our method into a RS visual odometry pipeline, in which the motion tracker could provide a good initial pose estimation.
>
>  **Q4: A Possible Direction for improvement is to regard the camera motion model as the parameters in the Nerf model and solve it during optimization. To this end, the proposed method can be more general to handle global and rolling shutter datasets. It will be a benefit for the community further.**
>
> A4: Our newly conducted experiments in A2-1 demonstrate that our method is already able to handle global shutter camera as a special case of RS camera. We do agree that integrate camera motion model as part of the NeRF model, could be a possible direction for further improvement.
>
>  **Q5: To sum up, due to the abovementioned concerns, I cannot give a positive rating to the proposed method in the current version.**
>
> A5: Providing those newly conducted experiments and explanations, we sincerely thank and wish the reviewer to reconsider the final rating.

---

> > ### Comment · Reviewer_Xdqv · 2023-11-21
> > **1**
> >
> > Thanks for the clarification.
> >
> > Q2 & Q4 :
> > I am not fully sure that the llff dataset is the global shutter one, although the author mentioned it was captured by cellphone.
> >
> > The result shows that the fixed camera motion modeling is inferior to the final PSNR and the other metrics.
> >
> > The proposed method with a fixed camera model only applied to the sequential-like dataset. Since the camera motion model is based on the timestamp, the proposed method should heuristically find an incremental sequence from the pose graph if not from the video sequence.
> >
> > The proposed method is slightly inferior to the baseline because the captured images in LLFF have a pattern from left to right, from top to down, and the filename also sorts the sequence of images. We can still find the T_0 to T_3 for the camera motion modeling. I think the proposed method will achieve a worse result if the global shutter images are shuffled. Thus, I don't agree with the claim that "Our method can handle global shutter camera as a special case of RS camera.", since the current version of camera motion model do not handle it at all.
> >
> > Furthermore, I agree with wLRe that the proposed method should compared with the 2-stage method as the baseline. While colmap is not the appropriate one since it a Another method without modeling rolling shutter. In SLAM/VO,  since these methods are sentitive to rolling shutter, thus some methods [1,2,3] modeling the rolling shutter during localazation and mapping.
> >
> > Alougth some of these methods are not open-sourced, you can still ask the aufor the poses on TUM-DS for comparasion because most of them have the experiments on TUM-DS.
> >
> > Q4:
> > Again, as I mentioned before, I am happy to adjust the rate to positive side if the author could modeling camera motion in a optimization way to handle both rolling shutter and global shutter images with any given pose graph. Since it will solve the most concerns mentioned by the reviewers(camera motion modeling, linear or cubic, etc.) While I cannot do so for this version of fixed camera motion model.
> >
> > **I am still open to discuss if the author could provide more results and clarification.**
> >
> > [1] D. Schubert et al. “Rolling-Shutter Modelling fdoes Visual-Inertial Odometry”. In: Internathe tional Conference on Intelligent Robots and Systems (IROS). Nov. 2019
> >
> > [2] Nan Yang et al. Challenges in Monocular Visual Odometry: Photometric Calibration, Motion Bias and Rolling Shutter Effect. 2018.
> >
> > [3] David Schubert et al. “Direct Sparse Odometry with Rolling Shutter”. In: Computer Vision ECCV 2018. Springer International Publishing, 2018, pp. 699–714.

---

> > > ### Comment · Reviewer_Xdqv · 2023-11-21
> > > **2**
> > >
> > > BTW, orb-slam3 also has the rolling shutter version.
> > >
> > > I also think the DROID-SLAM can be used as a baseline since it is the strongest baseline in most scenarios, which is far beyond the conventional slam/vo, like orb-slam.

---

> > > > ### Author Response · Authors · 2023-11-21
> > > >
> > > > **Q6: I also think the DROID-SLAM can be used as a baseline since it is the strongest baseline in most scenarios, which is far beyond the conventional slam/vo, like orb-slam**
> > > >
> > > > We are not a SLAM/Visual odometry pipeline. We are a dense rolling shutter bundle adjustment method, using NeRF as the underlying representation. Therefore, the more suitable baselines to compare are "RSBA-CVPR2012" and "NW-RSBA-CVPR2023". We actually reimplement the papers and compared against them. The experimental results show that our method works better than theirs, in Table 3, 8, 9, 10.

---

> > > ### Author Response · Authors · 2023-11-21
> > >
> > > **!!ADDITIONAL CLARIFICATION!! on "Q3: the proposed method should compared with the 2-stage method as the baseline."**
> > >
> > > Actually we also compared against traditional rolling shutter correction methods in our paper originally. In particular, we compare against a state-of-the-art non-learning based method "Zhuang etal., Rolling-shutter-aware differential SfM and image rectification, ICCV, 2017". It is newer than the method proposed by reviewer wLRe, which can be considered as a SOTA for traditional methods. The results are presented in Table 2, 5, 7, i.e. labled as "DiffSfM" and "NeRF + DiffSfM". The experimental results also demonstrate that our method performs better than it. The main reason that it does not perform well is due to the quality of the corrected images are not good and contain undesirable artifacts. We use their public implementation for the experiments.
> > >
> > > As the response to wLRe, we also compared against "learning based methods + NeRF" for our additional 2-stage experiments. The experimental results also demonstrate that our method is better than prior methods. The results are presented in Table 2, 5, 7 of the paper.

---

> ### Author Response · Authors · 2023-11-21
>
> **Q1: I am not fully sure that the llff dataset is the global shutter one, although the author mentioned it was captured by cellphone.**
>
> The images of LLFF dataset is captured while the camera is stationary, and they then move to the next position for capture. Even it is captured by a cell phone, it can still be considered as a global shutter image since the camera is static. There is no RS effects in this case.
>
> **Q2:  I don't agree with the claim that "Our method can handle global shutter camera as a special case of RS camera.", since the current version of camera motion model do not handle it at all.**
>
> Based on the response of Q1, we can claim that our method is able to handle global shutter image sequence, as the reviewer also agrees.
>
> In reality, we also conduct experiments with shuffled unorganzied rolling shutter dataset, the results are presented in Table 6. If it is an un-ordered dataset, we can easily re-arrage a single cubic-spline for the whole sequence, to multiple cubic-spline and each is defined for each image with only 4 control knots. The experimental results demonstrate that our method also can work for shuffled data.
>
> **Q3: the proposed method should compared with the 2-stage method as the baseline.**
>
> We actually compared against 2-stage method, i.e. deep learning RS correction + NeRF. The results are presented in Table 5. For the reasons not to run "traditional RS correction method + NeRF", we also response to wLRe. In particular, prior traditional methods usually have strong assumptions (i.e. planar, straight line is straight, etc), our method is already superior by analysing the formulation. Furthermore, they do not have their code released and we contact some authors for their implementation, but with no response. Therefore we did not compare against them.
>
> **Q4: Alougth some of these methods are not open-sourced, you can still ask the aufor the poses on TUM-DS for comparasion because most of them have the experiments on TUM-DS.**
>
> Regarding the pose estimation, we do implement two prior RS bundle adjustment methods (RSBA-CVPR2012, NW-RSBA-CVPR2023) by ourself for better evaluation, which are more related to ours setting. i.e. we are not a visual odometry pipeline. From Table 3, 8, 9, 10, we can see that our method is better than them. We did not compare against a weak pipeline. The NW-RSBA method is the newest state-of-the-art method from this year's CVPR.  In their work, they also compare against ORB-SLAM, and show that they are better than ORB-SLAM.
>
> On the other hand, our method focus more on novel view image synthesis, it is what none any prior RS visual odometry methods could achieve.
>
> **Q5:  I am happy to adjust the rate to positive side if the author could modeling camera motion in a optimization way to handle both rolling shutter and global shutter images with any given pose graph. Since it will solve the most concerns mentioned by the reviewers(camera motion modeling, linear or cubic, etc.) While I cannot do so for this version of fixed camera motion model.**
>
> Our current setting could already handle both global shutter images and rolling shutter images, for both sequential and shuffled datasets, as our response to Q1 & Q2. It might further improve the performance to do as what the reviewer suggested, but our current setting could already handle it.
>
> On the other hand, even though our method has no such limitation (to handle both RS and GS images), we are wondering the motivation to handle both rolling shutter and global shutter images with a single NeRF as the reviewer insists on, since we can easily switch between a global shutter NeRF and RS NeRF codebase for deployment.

---

### Official Review · Reviewer_UbiP · 2023-10-29

**Soundness:** 3 good
**Presentation:** 4 excellent
**Contribution:** 3 good
**Rating:** 8
**Confidence:** 5

**Summary:**

This paper proposes a method called Unrolling Shutter Bundle Adjusted Neural Radiance Fields (USB-NeRF) that corrects rolling shutter distortions and improves the accuracy of camera pose estimation. USB-NeRF models the physical image formation process of a rolling shutter camera and uses a bundle adjustment framework to optimize the camera poses and scene geometry. The technique unrolls the rolling shutter effect by modeling the exposure time of each pixel and correcting the time-varying motion of the camera. USB-NeRF also uses a neural radiance field to model the scene geometry and appearance, which allows for high-quality novel view synthesis. The paper includes tables and figures that show the quantitative and qualitative comparisons of USB-NeRF with other methods on synthetic and real-world datasets. The experimental results demonstrate that USB-NeRF achieves better performance compared to prior works in terms of RS effect removal, novel view image synthesis, and camera motion estimation.

**Strengths:**

1. This paper is very well written. I can easily understand the paper even though I'm not very familiar with the rolling-shutter camera.

2. The proposed method is simple yet effective, which uses the cubic B-Spline to interpolate between camera poses instead of linear interpolation.

3. The paper did exhaustive experiments to evaluate the effectiveness of their method on both the synthetic and real-world datasets. Though there is a lack of baseline methods for rolling-shutter NeRF, they compared with various methods that bundle-adjust rolling-shutter cameras.

**Weaknesses:**

1. I think the ATE (absolute trajectory error) in Table 3 is the same as the absolute translation error (I'm used to the term `translation` instead of `trajectory`). Therefore, only the translation errors are given and no rotation errors are provided. Moreover, the unit of the ATE is unclear (I think it is in meters).
2. The cubic B-Splines interpolation is suitable for complex camera trajectories, however, it can be worse than the linear interpolation method when the camera moves at a constant velocity.

**Questions:**

- From Fig. 6, BARF looks much worse than NeRF; and from other tables and figures, BARF performs almost the same as NeRF. Are there any explanations for this?

- Follow the question above. It is expected that BARF can fail under the rolling-shutter setting since each row of the image is recorded at different timestamps. My question is can we build a stronger baseline method that associates each row with a camera pose, then we can use BARF to optimize these camera poses and obtain better results? It can be time-consuming since an image can often have >400 rows, and then we have to optimize too many parameters. A simplified way is to split the image into R row blocks, where R <<< the width/height of that image. Then we have a simplified stronger baseline than BARF.

---

> ### Author Response · Authors · 2023-11-20
>
> We thank the reviewer for the acknowledgement of the effectiveness of our method and the paper writing quality. We make replies to the questions from the reviewer as follows:
>
>
> **Q1: There is a lack of baseline methods for rolling shutter NeRF.**
>
> A1: Since our method is the first NeRF-based method to model the rolling shutter effect, we are unable to conduct the experiment to compare against prior rolling shutter NeRF.
>
> **Q2: No rotation errors are provided and the unit of ATE is unclear.**
>
> A2: Yes, the ATE metric is mainly for absolute translation error and the unit is in meters. We have added the unit explanation in the table caption of the main paper to make it clearer.
>
> We re-computed the rotation error with the RPE metric (relative pose error) for rotation. The unit is in ``degrees per frame'' (i.e. $\Delta =1$). The results for both synthetic and real datasets are presented in the following table. Due to the space limit, we only report the average metrics here and the details for each sequence can refer to both Table 9 and Table 10 of the main paper. It demonstrates USB-NeRF also has a better rotation estimation accuracy than prior methods.
>
> |  |  COLMAP | BARF | RSBA | NW-RSBA | USB-NeRF-linear | USB-NeRF-cubic |
> |:--:|:--:|:--:|:--:|:--:|:--:|:--:|
> | Synthetic |  2.367$\pm$ 1.348 | 8.386$\pm$ 5.119 | 5.179$\pm$ 2.627 |  2.090$\pm$ 1.492  | 0.446$\pm$ 0.337 | __0.307__ $\pm$ 0.140 |
> | TUM-RS  | 0.393 $\pm$ 0.218 | 0.429 $\pm$ 0.430 | 2.095 $\pm$ 2.607 |  8.379 $\pm$ 9.928 | 0.283 $\pm$ 0.285 | __0.227__ $\pm$ 0.246 |
>
> **Q3: The cubic B-Splines interpolation is suitable for complex camera trajectories, however, it can be worse than the linear interpolation method when the camera moves at a constant velocity.**
>
> A3: Cubic B-Splines interpolation is indeed a bit worse than the linear interpolation method when the camera moves at a constant velocity, from the results of Table 1 in the paper. The PSNR metric becomes 0.25 dB worse, which we think is still tolerable.
>
> **Q4: BARF looks much worse than NeRF in Fig. 6.**
>
> A4: Thank you for pointing out. We are sorry that we put a wrong figure of BARF in Fig. 6. We corrected it in the main paper now. However, we find that BARF still performs a bit worse than NeRF on the **White Room** sequence. The main reason is that we need to optimize the camera poses of BARF for novel view synthesis experiment, to align the novel view image against the trained scene representation for evaluation. Therefore, the camera pose might converge to a wrong solution sometime if both the learned 3D representation and images are distorted. In contrary, NeRF does not require this since they fix the camera poses which are estimated from COLMAP.
>
> **Q5: Can we have a simplified stronger baseline via BARF by splitting the image into R row blocks?**
>
> A5: We conduct this interesting experiment with BARF. In particular, we split every image into both 3 blocks (160 lines per block), and optimize camera pose of each block separately via BARF. We also repeat the experiment with 5 blocks (96 lines per block). The experimental results are presented in following Table on synthetic datasets. It demonstrates that it can have improved performance on some sequences, but could also be worse for others. We think there are two possible reasons: 1) divide the whole image into chunks could indeed help since it suppress the distortions. However, each chunk could still have considerable distortion if the total number of chunks is small, which would affect the performance; 2) divide the whole image into too many chunks would lead to insufficient information of each chunk for individual pose optimization, which would also affect the final performance. Therefore, it would beneficial to model the RS effect into NeRF for better performance.
>
> | | |Blue Room| | |Living Room| | |White Room| | |Avg.| |
> |:--:|:--:|:--:|:--:|:--:|:--:|:--:|:--:|:--:|:--:|:--:|:--:|:--:|
> | | PSNR$\uparrow$ | SSIM$\uparrow$ | LPIPS$\downarrow$ | PSNR$\uparrow$  | SSIM$\uparrow$ | LPIPS$\downarrow$ | PSNR$\uparrow$  | SSIM$\uparrow$ | LPIPS$\downarrow$ | PSNR$\uparrow$ | SSIM$\uparrow$ | LPIPS$\downarrow$ |
> | BARF | 19.14 |  0.576 | 0.3446  | 21.48 |  0.690 | 0.3341  |  18.43  | 0.545 |  0.3837 | 19.68 | 0.604 |  0.3541 |
> | BARF*(3 blocks) |  19.65  | 0.621 |  0.3552 |  19.74  | 0.682 |  0.3565 |  18.47  | 0.532 |  0.3546 | 19.29 | 0.611 |  0.355  |
> | BARF*(5 blocks) | 20.12  | 0.634 |  0.2429 |  17.97  | 0.650 |  0.4310 | 19.04  | 0.557 |  0.3354 | 19.04 | 0.614 |  0.3364 |
> | USB-NeRF (ours) | __31.85__ | __0.909__  | __0.0573__ | __34.89__ | __0.939__  | __0.0415__ | __30.57__ | __0.892__  | __0.0576__ | __32.44__ | __0.914__ | __0.0521__ |

---

> > ### Comment · Reviewer_UbiP · 2023-11-23
> > **Thanks for your reply**
> >
> > Thanks to the authors for the rebuttal. All of my questions are answered. Therefore, I decided to keep my score as accepted.

---

### Official Review · Reviewer_wLRe · 2023-10-29

**Soundness:** 2 fair
**Presentation:** 2 fair
**Contribution:** 2 fair
**Rating:** 3
**Confidence:** 4

**Summary:**

The paper proposes a method to handle rolling shutter problem in NeRF reconstruction. In particular, it proposes a method to rectify the input images caused by rolling shutter followed by NeRF reconstruction. Experimental results show that the proposed method can handle distortion caused by rolling shutter effectively.

**Strengths:**

The proposed method is evaluated on both synthetic and real world dataset and demonstrated the improvement of reconstruction with and without the rolling shutter correction.

**Weaknesses:**

I am not fully convinced that using rolling shutter camera is an effective way to capture a NeRF model. There is actually no motivation/benefits to use rolling shutter camera to capture a NeRF model.

Considering the case that using rolling shutter camera is necessary, the proposed solution is just a simple two-step approach with first rolling shutter correction followed by NeRF reconstruction. I do not see any connection between rolling shutter correction and NeRF reconstruction in the proposed method. Since there is no connection, the rolling shutter correction method is just a standard method which estimate motion trajectory followed by rectification. From the formulation, I do not see any technical novelty.

**Questions:**

Please try to correct me if I have made any mistakes on the evaluation of this submission.

---

> ### Author Response · Authors · 2023-11-20
>
> We thank the reviewer for the effort spent in reviewing our paper and the acknowledgement of the improved performance with rolling shutter modeling. We make replies to the questions from the reviewer as follows:
>
> **Q1: I am not fully convinced that using rolling shutter camera is an effective way to capture a NeRF model.**
>
> A1: As the comment from reviewer CtJa, most consumer products (e.g. GoPro camera, DSLR camera and iPhone 14 Pro) still use rolling shutter cameras for photo capture and video recording nowadays. Since we usually capture images in a slow motion, the RS effect is often negligible. However, there are still obvious RS distortions for certain cases if the camera undergoes fast motion, e.g. recording videos with a GoPro camera carried by a flying drone, which is a common setup for most hobbyists. Therefore, it is valuable to model RS effect for NeRF reconstruction.
>
> **Q2: The proposed method is a simple two-step approach, which lacks novelty.**
>
> A2: Our method **is not a simple two-step approach**, which conducts rolling shutter correction first and then run normal NeRF reconstruction. Instead, our method **is a single-step method**. In particular, we integrate the RS image formation model into the framework of NeRF, and use RS images to directly reconstruct the true underlying 3D scene representation with NeRF, which has not been done by any prior work. Once the model is trained, corrected global shutter images can be rendered from NeRF to achieve rolling shutter effect correction.

---

> > ### Comment · Reviewer_wLRe · 2023-11-20
> >
> > Thanks authors for the feedback.
> >
> > 1) Regarding the high speed motion, none of your examples contain moving objects. The only motion is camera motion. When capturing a 3D static scene on purpose, I am not convinced that a user would purposely move the camera in high speed and it is an uncommon scenario to have such rolling shutter effects unless they are captured on purpose.
> >
> > 2)  I apologize if I have made a mistake in accessing the proposed method. However, I am puzzle why such simple 2-step approach do not work on this application scenario? Although you have provided comparisons with RS corrections based on deep learning, there are many traditional rolling shutter correction methods that can be applied in this application scenario and they do not require training data. In particular:
> >
> > Rolling Shutter Bundle Adjustment, CVPR 2012
> > Global Optimization of Object Pose and Motion from a Single Rolling Shutter Image with Automatic 2D-3D Matching, ECCV 2012
> > Calibration-free rolling shutter removal, ICCP 2012
> >
> > The above traditional methods should also fit well to the NeRF capturing scenario since it is not just based on 2D, but also 3D SfM. Note that you have also cited the CVPR 2012 paper (Hedborget al.,2012) and claimed that their model does not work. However, there is no such comparison in the paper. The major difference between the proposed method and the CVPR 2012 work is that CVPR 2012 use piecewise linear motion model, and the proposed method uses cubic B-Spline motion model to fit the camera trajetory. Indeed, I do not see much differences in such camera motion modelling, even there is, the differences would be subtle.
> >
> > 3) This paper, while packaging for NeRF reconstruction, is closer to rolling shutter correction. However, it is disappointed that the paper does not done enough in literature review in rolling shutter correction, and many methods in rolling shutter correction are ignored in the comparisons.

---

> > > ### Author Response · Authors · 2023-11-21
> > >
> > > **!!ADDITIONAL CLARIFICATION!! on "Q2: Although you have provided comparisons with RS corrections based on deep learning, there are many traditional rolling shutter correction methods that can be applied in this application scenario and they do not require training data."**
> > >
> > > Actually we also compared against traditional rolling shutter correction methods in our paper originally. In particular, we compare against a state-of-the-art non-learning based method "Zhuang etal., Rolling-shutter-aware differential SfM and image rectification, ICCV, 2017". It is newer than the method proposed by the reviewer, which can be considered as a SOTA for traditional methods. The results are presented in Table 2, 5, 7, i.e. labled as "DiffSfM" and "NeRF + DiffSfM". The experimental results also demonstrate that our method performs better than it. The main reason that it does not perform well is due to the quality of the corrected images are not good and contain undesirable artifacts. We use their public implementation for the experiments.

---

> ### Author Response · Authors · 2023-11-21
>
> We thank the reviewer for the prompt feedback on our responses! We make replies to the questions from the reviewer as follows:
>
> **Q1: No examples contain moving objects.**
>
> Our method is currently not modeled to work for dynamic environments. However, our method is the first to train NeRF with rolling shutter images of static environment. We propose to divide the solution into two steps: handle static environment first, and leave the solution for dynamic environments as our next work. The capability to handle static environments would already be useful for many application scenarios. On the other hand, to handle dynamic environments with NeRF itself is also an on-going research.
>
> **Q2: I am not convinced that a user would purposely move the camera in high speed and it is an uncommon scenario to have such rolling shutter effects unless they are captured on purpose.**
>
> The rolling shutter effect is not negligible for certain scenarios and not purposely to have it. For example, to reconstruct a 3D scene with a GoPro camera carried by a fast flying drone. It would affect the users experience to instruct them to fly the quad-copter slowly, due to our software cannot handle RS distortion. It is also not practical to ask them to replace their GoPro camera with a more expensive global shutter camera, which could also be bulky for the drone to carry.
>
> If we could have the solution without requiring any changes for the end consumers, we think it would still be beneficial to have it.
>
> **Q3: I am puzzle why such simple 2-step approach do not work on this application scenario? Although you have provided comparisons with RS corrections based on deep learning, there are many traditional rolling shutter correction methods that can be applied in this application scenario and they do not require training data. In particular...**
>
> We will reply from following perspectives:
>
> 1) A simple 2-step approach would heavily rely on the performance of the RS correction step. Instead of doing this, we did not do explicit RS correction and integrate the image formation model into NeRF, such that NeRF can be learned from RS images directly; The experimental results demonstrate our method can deliver good performance;
>
> 2) We do compare against such 2-step approach in our paper, as the reviewer also agrees, except that we do not compare against traditional approach + NeRF; Due to the generalization performance, deep learning based 2-step approach does not work well as what we demonstrate in our experiments;
>
> 3) The three traditional methods suggested by the reviewer have no implementation released to the public. Even though, we implemented the "Rolling Shutter Bundle Adjustment, CVPR 2012'' paper by ourself for comparisons. The experimental results are presented in Table 3, Table 8, Table 9 and Table 10. They do fail on some sequences. The shortcut "RSBA'' is for this work and we explained it in the "Evaluation metrics'' section on page 7;
>
>    We also implement another more recent traditional approach ``NW-RSBA, CVPR 2023'' for better evaluations;
>
> 4) The method from "Rolling Shutter Bundle Adjustment, CVPR 2012'' does not support rolling shutter correction. It is designed to do sparse bundle adjustment with rolling shutter images, and there is no dense rolling shutter correction involved;
>
> 5) The method from "Global Optimization of Object Pose and Motion from a Single Rolling Shutter Image with Automatic 2D-3D Matching, ECCV 2012'' does not support rolling shutter correction either. It is designed to estimate the pose of a moving object relative to the camera. There is no dense rolling shutter correction involved;
>
> 6) The method from "Calibration-free rolling shutter removal, ICCP 2012'' does consider rolling shutter correction. However, they exploit homography to formulate the problem, which would fail if the scene is composed of layers with large differences in depth. In contrary, our method does not have this limitation, and can handle general scenarios, as the results in Figure 10 demonstrate.
>
> By considering all these factors, we re-implement two traditional methods "RSBA, CVPR 2012'' and "NW-RSBA, CVPR 2023'' for evaluations. Traditional methods, such as "Calibration-free rolling shutter removal, ICCP 2012'' usually have strong assumptions and cannot deliver good RS correction results for general scenarios. We therefore did not re-implement and compare against them for the 2-step approaches. Our method does not have such limitations.

---

> ### Author Response · Authors · 2023-11-21
>
> **Q4: Note that you have also cited the CVPR 2012 paper (Hedborget al.,2012) and claimed that their model does not work. However, there is no such comparison in the paper.**
>
> We do have this comparison in the paper, the results are presented in Table 3, Table 8, Table 9 and Table 10. The "RSBA'' shortcut represent the paper and we explained it in the "Evaluation metrics'' section on page 7. Since RSBA is used to estimate the camera pose and reconstruct 3D points (i.e. Sparse bundle adjustment), we therefore only compare against it in terms of the pose accuracy and they do fail on some sequences.
>
> **Q5: it is disappointed that the paper does not done enough in literature review in rolling shutter correction, and many methods in rolling shutter correction are ignored in the comparisons.**
>
> We included more related works as presented in the related work section on Page 3. Since there are many prior methods, we are unable to include every related paper, and only present several representative methods.
>
> Based on previous responses, we compare against traditional methods, RSBA-CVPR2012, NW-RSBA-CVPR2023, even though they do not have their implementations released and we implemented them by ourselves for better evaluations.
>
> For other traditional methods, which usually do not release their code to the public, such as "Calibration-free rolling shutter removal, ICCP 2012'', we did not re-implement and compare against it since we can already see the advantages of our method over theirs, based on the limitations of their formulation (i.e. usually require strong assumptions, while our method does not have such limitation).
>
> For recent deep learning based methods, we conduct the experiments and the experimental results are presented in Table 2, Table 5, Table 7.
>
> **Providing those clarifications, we sincerely wish and thank the reviewer could re-consider the final rating.**

---

> ### Comment · Reviewer_wLRe · 2023-11-21
>
> Regarding the comments on moving objects, I am not requesting the authors to provide results with moving objects. Instead, I was questioning the motivations and the necessity of using a rolling shutter camera with fast camera motion to capture a static scene. The rebuttal claimed that GoPro carried by a flying drone is one of the target scenario, but none of the examples were captured under this setting. Again, if it is a static scene, and the user has full controls on capturing device and speed, I do not see a convincing reason to capture a scene with a fast moving rolling shutter camera.
>
> Regarding whether it's one-step/two-step approach, the most important part is getting the correct ray projection and the corresponding observations. In your proposed method, you embedded them into the NeRF setting, but it is also applicable that one can first correct the rolling shutter effects followed by NeRF reconstruction. Note that you have also mentioned "Furthermore, our algorithm can also be used to recover high-fidelity high frame-rate global shutter video from a sequence of RS images." in your abstract. The key question is what are the benefits that the one-step approach brings over the two-step approach if the ray projection and the corresponding observations are already correctly determinate. I believe there is difference when the rolling shutter effects are severe, but the differences would be subtle if the rolling shutter effects are minor. Again, it goes back to the motivation why one has to use fast moving rolling shutter camera to capture a 3D static scene.
>
> I asked for the additional comparisons because the major technical challenge in this problem setting is the rolling shutter correction, and the methods you have compared are not targeted for the same application scenario. The one I mentioned in CVPR 2012 which corrects the bundle adjustment, i.e. ray projection and the corresponding observations, for 3D SfM is closely related to your work. As I have already mentioned in the responses, the major differences are the camera motion model, theirs is piecewise linear motion model, and yours is cubic B-spline model. Although you have provided quantitative comparisons, are the improvements of using cubic B-spline model over piecewise linear motion model significant? How good/bad does these models fit to the real-world scenario? Of course, if we only look at the quantitative number, one can easily get a better score using synthetic examples since the camera path can be generated to overfit one model over the others. But, the real question is, again, how well does these parametric motion model modelling the real world camera motions, e.g. the drone flying motion if this is the target scenario. Since both are not accurate motion model, I would have to say their results are going to be similar.

---

> ### Author Response · Authors · 2023-11-22
>
> We thank the reviewer for the prompt feedback and interaction with us. We make replies to the questions from the reviewer as follows.
>
> **Q1: Instead, I was questioning the motivations and the necessity of using a rolling shutter camera with fast camera motion to capture a static scene. The rebuttal claimed that GoPro carried by a flying drone is one of the target scenario, but none of the examples were captured under this setting. Again, if it is a static scene, and the user has full controls on capturing device and speed, I do not see a convincing reason to capture a scene with a fast moving rolling shutter camera.**
>
> Our responses are in following two perspectives:
>
> 1): It is currently impractical for us to purchase a drone with a GoPro camera to capture the data due to the available time left. However, we tried to simulate the setting as much as possible. For example, the results presented in Figure 11 are captured on a moving tram with a GoPro HERO6. There is no much difference to carry the GoPro camera with a tram or a drone. Furthermore, we also cannot control the speed of a public tram, so it is not on purposely captured either.
>
> We can do ask them to control the flying speed of the camera, however, it affects the user experience to instruct them "you have to fly slowly due to the RS effect". If they are hobbyists and want to simply do NeRF reconstruction using a video captured during their fun flight (in which they usually aim for fast flight just for fun), they would not choose to control the speed and would choose to throw away the software and choose the one which could satisfy them.
>
> We simply choose the "GoPro carried by a Drone" as an example. There are also many other example scenarios. For example, the video capture on a moving tram/bus. If we can choose to have this possibility to have a RS NeRF to better handle these scenarios, we think there is no harm to have it.
>
> 2): If the reviewer still disagrees with the above scenarios on the motivation to do NeRF with a rolling shutter camera, our method can also be considered as a dense bundle adjustment method (with NeRF as the underlying 3D representation) for the contribution, which also delivers better performance than prior state-of-the-art methods on rolling shutter aware bundle adjustment, i.e. RSBA-CVPR-2012 and NW-RSBA-CVPR-2023 methods. We also choose to use a public TUM-RS dataset for evaluation to avoid the suspect of using overfitted synthetic datasets. In reality, the trajectories used to create our synthetic datasets are also real trajectories adopted from the ETH3D dataset. It is not on purposely created trajectories.
>
> **Q2: Regarding whether it's one-step/two-step approach... I believe there is difference when the rolling shutter effects are severe, but the differences would be subtle if the rolling shutter effects are minor. Again, it goes back to the motivation why one has to use fast moving rolling shutter camera to capture a 3D static scene."**
>
> 1): To recover high-fidelity high frame-rate global shutter video is based on that the NeRF is already trained with RS images, not the vice verse. It is another advantage of our method over prior one.
>
> 2): Based on the response, I think the reviewer is now convinced that two-step approach would not work well for images with large RS effects.
>
> 3): For learning based RS correction method, the difficulty to deliver consistent high-quality global shutter images are not merely caused by the large RS effects. It is mainly caused by the poor generalization performance. Even the RS effect is not large, they might still not be able to generalize well on images from a different distribution domain. Traditional methods usually require strong assumptions for the ease of formulation, it is also fragile to deliver consistent high-quality global shutter images once the assumption is violated. Our method has no such limitations.
>
> 4): Furthermore, the dataset that we used for evaluation also does not exhibit very large RS effect. For example, the images presented in Figure 1 of the paper have similar level of RS effects, as what we captured on a city tram, which does not move that fast usually.
>
> The experimental results also demonstrate that two-stage approach performs worse than our method, even with these minor realistic RS effects.
>
> 5): To do NeRF reconstruction with captured video on a moving city tram/bus, is also a potential real application scenario, which is not on purposely captured either.
>
> If we could have a robust non-fragile method to be able to handle different levels of RS effect in one-go, it would be beneficial to have it. Furthermore, our formulation is also novel and has no generalization problem.

---

> ### Author Response · Authors · 2023-11-22
>
> **Q3: I asked for the additional comparisons because the major technical challenge in this problem setting is the rolling shutter correction, and the methods you have compared are not targeted for the same application scenario...Of course, if we only look at the quantitative number, one can easily get a better score using synthetic examples since the camera path can be generated to overfit one model over the others...**
>
> 1): Our method has two main purposes: a) to do rolling shutter correction under the NeRF framework; b) to do dense bundle adjustment for more accurate pose estimation;
>
> 2): Regarding rolling shutter correction, we have compared against prior state-of-the-art deep learning (i.e. DSUN, SUNet, RSSR and CVR) and traditional methods (i.e. DiffSfM-ICCV-2017). Regarding bundle adjustment, we also have compared against prior SOTA methods, i.e. RSBA-CVPR-2012 and NW-RSBA-CVPR-2023. Furthermore, we also compared against NeRF and BARF.
>
> We could not agree with the reviewer that the methods we have compared are not targeted for the same application scenarios.
>
> 3): Our method does have similarity in using cubic-spline or linear interpolation with RSBA-CVPR2012. However, there are other key differences between them, which is that RSBA-CVPR2012 relies on sparse keypoints while our method exploits the full image for optimization with photometric error. Furthermore, the underlying 3D representation is also different, i.e. RSBA-2012 exploits sparse point cloud, while ours exploit a neural network for the representation.
>
> 4): We also did not use synthetic datasets for the comparisons solely. We also exploit public real dataset from TUM-RS for comparisons on both pose estimation and rolling shutter correction. Furthermore, the trajectories of our synthetic datasets are also adopted from real dataset, i.e. ETH3D, which are not created on purposely to overfit the motion model. These experimental results demonstrate that our method delivers better performance than prior work.

---

### Official Review · Reviewer_CtJa · 2023-10-30

**Soundness:** 4 excellent
**Presentation:** 4 excellent
**Contribution:** 3 good
**Rating:** 8
**Confidence:** 4

**Summary:**

The paper proposes a method to model rolling shutter effect during NeRF training. The main idea is to model camera trajectory with a B-spline, which allows time interpolation so that each scanline in an image can be associated with a more accurate camera pose at the moment of when the line of pixel is captured.

**Strengths:**

1. The paper is well written and easy to follow.
2. The proposed idea is novel and technically solid.
3. The evaluation is convincing and supports the main contribution well.

**Weaknesses:**

I don’t see any major weakness.

It would be interesting to see more analysis/discussions on how much is the performance gap when modelling rolling shutter effect vs not modelling with datasets from modern DSLR cameras and smart phone cameras.

**Questions:**

See the weakness section.

---

> ### Author Response · Authors · 2023-11-20
>
> We thank the reviewer for the acknowledgement of the technical novelty and soundness of our method. We make replies to the questions from the reviewer as follows:
>
> We captured 5 additional sequences using GoPro HERO6 Black, Canon camera (EOS M3), and iPhone 14 Pro, and made more analysis on real-world datasets. All rolling shutter images are captured at the frequency of 30Hz. We find that the captured images with modern cameras still exhibit obvious RS effects if they undergo fast motions, which can easily happen for certain scenarios, e.g. GoPro carried by a flying drone, which is a common set-up for most hobbyists or taking photos on a moving vehicle.
>
>
> Since it is hardly to capture corresponding GS images, we only make qualitative comparisons on these datasets. Please refer to Figure 10, Figure 11, and Figure 12 in Appendix A.3 of the main paper for the experimental results on real-world datasets.
>
>
> The experimental results demonstrate that: 1) both NeRF and BARF fail to correct the RS effect distortion; 2) both NeRF and BARF produce significant artifacts if the camera motion is not in a constant direction (e.g. the camera is moving forward and then backward); 3) our method corrects the RS effect distortion with no artifact; 4) it is still valuable to model RS effect for certain scenarios with modern cameras.

---

### Author Response · Authors · 2023-11-23
**Summary of the comments and discussions**

Dear area chair, reviewers,

We sincerely thank for your valuable comments, prompt feedback and discussions on our work. Since the discussion period is going to close, we make a brief summary as follows, based on the comments and discussions we had together recently.

**We think we have reached following consensus:**

1. Our method is novel and technically solid, i.e. we are the first to propose to do bundle adjustment with NeRF as the underlying 3D representation for rolling shutter images. Our method is able to deliver:
     - higher-fidelity corrected global shutter images than prior methods;
     - more accurate and stable pose estimations compared to prior RS bundle adjustment methods;
     - high fidelity novel view images from the learned NeRF representation (with RS images as the input).

2. The experimental results support our main contributions.
     - regarding RS correction, we compared against prior state-of-the-art deep learning (i.e. DSUN, SUNet, RSSR and CVR) and traditional RS correction methods (i.e. DiffSfM-ICCV-2017), in Table 2, 7;
     - regarding bundle adjustment, we compared against prior SOTA RS bundle adjustment methods, i.e. RSBA-CVPR-2012 and NW-RSBA-CVPR-2023, in Table 3, 8, 9, 10;
     - regarding novel view image synthesis, we compared against NeRF, BARF, NeRF+DiffSfM, NeRF+DSUN, NeRF+SUNet, NeRF+RSSR and NeRF+CVR, in Table 5;
     - we exploit both synthetic and real datasets (both public and private) for evaluations. The experimental results demonstrate that our method delivers better performance than prior SOTA methods. To better illustrate the advantages of our method, we also create a new video with real datasets captured by a Canon camera and iPhone 14 Pro on moving vehicles (i.e. city tram and bus). The video can be found from our supplementary or following anonymous link [https://youtu.be/XJ5w6_wifFo](https://youtu.be/XJ5w6_wifFo). Additional qualitative results can be also found in Fig. 1, 4, 5, 8, 9, 10, 11, 12, 13 and 14.

**We still have following concerns from wLRe and Xdqv, which prevent them from moving to the positive side:**

1. Reviewer wLRe is not convinced if we really need to do NeRF with RS images.

   *Summaries of our response:*
    - as mentioned previously, our method targets for rolling shutter aware bundle adjustment with NeRF as the underlying 3D representation. NeRF reconstruction is not the sole purpose of our method, and is only "one of the outcomes". We mainly exploit NeRF to have a better 3D scene representation. As outcomes, our method delivers high-quality RS correction, accurate pose refinement and high-fidelity novel view image synthesis. For each outcome, we all have thorough experiments to support that our method delivers superior performance over prior SOTA methods;
    - most consumer products (e.g. GoPro, DSLR cameras, iPhone 14 Pro etc.) still exploit RS cameras as their main imaging/video recording sensors nowadays. RS distortion is still common for certain scenarios, e.g. capturing video on a moving bus as what we presented in Figure 12;
     - furthermore, the datasets used for our evaluation actually does not exhibit on purposely severe RS effects, as presented in Figure 1 of the paper. The level of RS distortion is actually similar to that of our real datasets captured on a moving city bus/tram. Even with images exhibiting realistic level of RS effect, our method also deliver superior performance than prior methods, i.e. both learning-based and traditional methods.

2. Reviewer Xdqv thinks we need to model the camera motion in an optimization way as part of NeRF to handle both rolling shutter and global shutter images with any given pose graph.

     *Summaries of our response:*
     - as mentioned above, our method aims for RS-aware bundle adjustment with NeRF as the underlying 3D representation. Thorough experimental results demonstrate that our method is able to deliver high quality RS correction, accurate pose estimation and high fidelity novel view image synthesis, compared to prior works. The ability to deal with global shutter images is not the focus of our work;
    - even GS NeRF is not the focus of our work, we still present additional experimental results to Xdqv to demonstrate that our method is already able to handle sequential global shutter images, by using the LLFF dataset which is commonly used by the community. Though the images are captured by a cell phone, there is no motion (thus no RS effect) during the image capture. So it can be considered as global shutter images. The performance is comparable as NeRF and BARF on those images, or even slightly better for some sequences;
    - we also present experimental results that our method can be easily adapted for shuffled un-organzied RS dataset in Table 6 of the paper. It has no restriction to be applied for shuffled un-organized global shutter images, by considering GS image as a special case of RS image.

---

### Meta-Review · Area_Chair_Vsqt · 2023-12-07

**Metareview:**

The submission received mixed reviews. The reviewers appreciate the simplicity of the method and strong results demonstrated. The major concerns specifically raised by wLRe and Xdqv were:
- Questionable motivation (why a static scene should be captured with a fast-moving rolling shutter camera)
- Impractical motion models used, as well as in the data generated from the synthetic scenes
- Lack of comparisons against two-stage methods (RS correction then NeRF) and traditional techniques

The AC carefully read through the paper, the reviewers' comments, the authors' rebuttal and the discussions. Part of the concerns were caused by misunderstanding of the method, which the AC believes the authors have addressed in the discussions. Although the AC agrees with wLRe that there are rarely applications to capture static scenes with commodity sensor devices by modeling rolling shutter effects, the AC believes that technology advancements may in turn enable new applications, and there is still merit in a more thorough understanding of the technologies and their limitations. The AC also believes that the authors have adequately addressed issues regarding comparisons. On the other hand, the AC agrees that issues regarding motion models, both for generality and for practical use cases, should be further investigated.

Overall, the AC believes the merits outweigh the limitations in general, and thus the AC recommends acceptance.

**Justification For Why Not Higher Score:**

The applications of the work, as well as the overall scope, is more limited.

**Justification For Why Not Lower Score:**

The paper address a new problem of modeling NeRF from rolling shutter image captures. This would be of interest to the community more on the theoretical side.

---

### Decision · Program_Chairs · 2024-01-16

Accept (poster)